# The aging transcriptome and cellular landscape of the human lung in relation to SARS-CoV-2

Ryan D. Chow [ID] [1,2,3,4], Medha Majety[1,2,3,5] & Sidi Chen [ID] [1,2,3,4,6,7,8,9,10,11,12,13,14 ✉]

Age is a major risk factor for severe coronavirus disease-2019 (COVID-19). Here, we interrogate the transcriptional features and cellular landscape of the aging human lung. By intersecting these age-associated changes with experimental data on SARS-CoV-2, we identify several factors that may contribute to the heightened severity of COVID-19 in older populations. The aging lung is transcriptionally characterized by increased cell adhesion and stress responses, with reduced mitochondria and cellular replication. Deconvolution analysis reveals that the proportions of alveolar type 2 cells, proliferating basal cells, goblet cells, and proliferating natural killer/T cells decrease with age, whereas alveolar fibroblasts, pericytes, airway smooth muscle cells, endothelial cells and *IGSF21*[+] dendritic cells increase with age. Several age-associated genes directly interact with the SARS-CoV-2 proteome. Age-associated genes are also dysregulated by SARS-CoV-2 infection in vitro and in patients with severe COVID-19. These analyses illuminate avenues for further studies on the relationship between age and COVID-19.

[1] Department of Genetics, Yale University School of Medicine, New Haven, CT, USA. [2] Systems Biology Institute, Yale University, West Haven, CT, USA. [3] Center for Cancer Systems Biology, Yale University, West Haven, CT, USA. [4] M.D.-Ph.D. Program, Yale University, New Haven, CT, USA. [5] The College, Yale University, New Haven, CT 06520, USA. [6] Immunobiology Program, Yale University, New Haven, CT, USA. [7] Molecular Cell Biology, Genetics, and Development Program, Yale University, New Haven, CT, USA. [8] Combined Program in the Biological and Biomedical Sciences, Yale University, New Haven, CT, USA. [9] Yale Comprehensive Cancer Center, Yale University School of Medicine, New Haven, CT, USA. [10] Department of Neurosurgery, Yale University School of Medicine, New Haven, CT, USA. [11] Yale Stem Cell Center, Yale University School of Medicine, New Haven, CT, USA. [12] Yale Liver Center, Yale University School of Medicine, New Haven, CT, USA. [13] Yale Center for Biomedical Data Science, Yale University School of Medicine, New Haven, CT, USA. [14] Yale Center for RNA Science and Medicine, Yale University School of Medicine, New Haven, CT, USA. ✉email: sidi.chen@yale.edu

A ge is one of the strongest risk factors for severe outcomes among patients with COVID-19[1–6]. In the OpenSAFELY cohort of over 17 million patients in England, people over the age of 80 had more than twenty times the risk of COVID-19-related death compared to people aged 50–59 years old[3]. The OpenSAFELY study further noted an approximate log–linear relationship between risk of COVID-19-related death and age, indicating that the risk of COVID-19 mortality progressively increases over the spectrum of the human lifespan[3]. Although the OpenSAFELY cohort excluded patients under the age of 18, other lines of evidence have extended these conclusions to younger populations. For instance, the clinical manifestations of infection in children (<18 years old) are generally less severe than in adults[7–9]. With the exception of infants and younger children (<1 year old and 1–5 years old, respectively), many children are asymptomatic or experience mild illness[10,11]. Collectively, these observations indicate a strong association between age and COVID-19 morbidity and mortality. However, it must be emphasized that younger patients can still frequently contract the disease, possibly causing serious symptoms such as multisystem inflammatory syndrome[12], leading to hospitalization, intensive care unit admission, or death. While the effects of age on COVID-19 are likely to be multifactorial, involving a complex blend of systemic and local factors, we hypothesized that tissue-intrinsic changes that occur with aging may offer valuable clues.

In this study, we investigate the transcriptomic features and cellular landscape of the aging human lung in relation to SARS-CoV-2. We find that the aging lung is transcriptionally characterized by increased cell adhesion and heightened stress responses, along with reduced mitochondria and diminished cellular replication. Through deconvolution analysis, we identify numerous age-associated alterations in the cellular composition of the lung, including cell types that are implicated in host responses to SARS-CoV-2. We then cross-reference these age-associated factors with recent experimental data on host interactions with SARS-CoV-2, revealing that age-associated genes interact with the SARS-CoV-2 proteome and are also commonly dysregulated by SARS-CoV-2 infection in vitro. Notably, these findings are recapitulated in patients with severe COVID-19. This study illuminates potential mechanisms by which age influences the clinical manifestations of SARS-CoV-2 infection, pinpointing specific characteristics of the aging human lung that may contribute to the heightened severity of COVID-19 in older populations.

## Results

We focused our analysis on the Genotype-Tissue Expression (GTEx) project[13,14], a comprehensive public resource of gene expression profiles from non-diseased tissue sites. As the lung is the primary organ affected by COVID-19, here we specifically analyzed lung RNA-seq transcriptomes from donors of varying ages (21–70 years old) (Fig. 1a). A total of 578 lung RNA-seq profiles from 578 different donors were compiled, of which 31.66% were from women (Supplementary Data 1). For downstream analyses involving clinical covariates, we only included lung RNA-seq profiles from donors with complete annotations for sex, smoking status, and Hardy scale (*n* = 561).

**Factors associated with expression of SARS-CoV-2 entry factors.** An initial hypothesis for why SARS-CoV-2 differentially affects patients of varying ages is that the expression of host factors essential for SARS-CoV-2 infection may increase with aging[15–17]. To assess this possibility, we examined the gene expression of *ACE2* (Supplementary Data 2), which encodes the

protein angiotensin-converting enzyme 2 that is coopted as the host receptor for SARS-CoV-2[9,18–21]. Through multivariable regression analysis of several clinical factors, we observed a significant association between age and *ACE2* expression (estimated coefficient [95% confidence intervals] = 0.0061 [0.0026–0.0096], *p* = 0.00064) (Supplementary Fig. 1a, Supplementary Data 3). Of note, a major factor influencing *ACE2* expression was the Hardy scale, which describes the timescale of the circumstances surrounding a donor's death. The Hardy scale has been shown to have significant impacts on gene expression in the GTEx dataset, given its association with postmortem ischemic time[22]. With Hardy scale 1 (violent and fast death) as the reference, donors with Hardy scale 0 (on ventilator prior to death) had significantly higher *ACE2* expression (estimated coefficient = 0.5173 [0.3424–0.6922], $p = 1.07 \times 10^{-8}$), as previously reported[23,24]. Examining only the lung samples from donors that were not on a ventilator prior to death, *ACE2* expression increased with age (Supplementary Fig. 1b).

While ACE2 is the direct cell surface receptor for SARS-CoV-2, transmembrane serine protease 2 (TMPRSS2) and cathepsin L (CTSL) have been demonstrated to facilitate SARS-CoV-2 infection by priming the spike protein for host cell entry[20]. Expression of the corresponding genes *TMPRSS2* and *CTSL* were not significantly associated with age in the multivariable regression model, nor after excluding patients that were on a ventilator prior to death (Supplementary Fig. 1c–e). Notably, biological sex was not associated with the expression of *ACE2*, *TMPRSS2*, or *CTSL*, though it has been observed that males are more likely to be affected by COVID-19 than females[3,25–27]. One important limitation of these analyses is that the expression of SARS-CoV-2 entry factors is cell type-specific[15,28–31] (Supplementary Fig. 2a–c, Supplementary Fig. 3a–c, Supplementary Data 4, Supplementary Data 5). Transcriptional changes associated with different clinical features (e.g., aging, sex, or smoking status) may only occur in specific cell types, and could therefore be obscured in analyses of bulk transcriptomes[15,16].

**Identification of age-associated genes in human lung.** As indicated by our analysis of SARS-CoV-2 entry factors, in order to systematically identify age-associated genes in the human lung, it is important to account for clinical variables that may influence lung gene expression independently of age. Using a likelihood-ratio test[32] controlling for sex, smoking status, and Hardy scale, we pinpointed the genes for which age significantly impacts their expression (adjusted *p* < 0.05, Supplementary Data 6). We identified two clusters of genes in which their expression progressively changes with age (Fig. 1b, Supplementary Data 7).

Totally, 436 genes were found to increase in expression with age, while 346 genes decreased in expression with age (hereafter referred to as Age-Up and Age-Down genes, respectively). Gene ontology and pathway analysis of Age-Up genes revealed significant enrichment for cadherins, cell adhesion, stress response, extracellular matrix, and heat shock proteins, in addition to other pathways (Fig. 1c, Supplementary Data 8). These findings are consistent with the age-associated architectural changes in the lung[33,34], as well as the induction of stress pathways with aging[35]. In contrast, Age-Down genes were significantly enriched for cell cycle, DNA damage/repair, and mitochondrial factors, among other pathways (Fig. 1d, Supplementary Data 9), which is in line with prior observations of progressive mitochondrial dysfunction and loss of regenerative capacity with aging[36–39]. Notably, a recent computational model of intracellular RNA localization predicted that the SARS-CoV-2 genome and associated transcripts are enriched in the mitochondrial matrix[40]. Age-associated alterations in mitochondrial

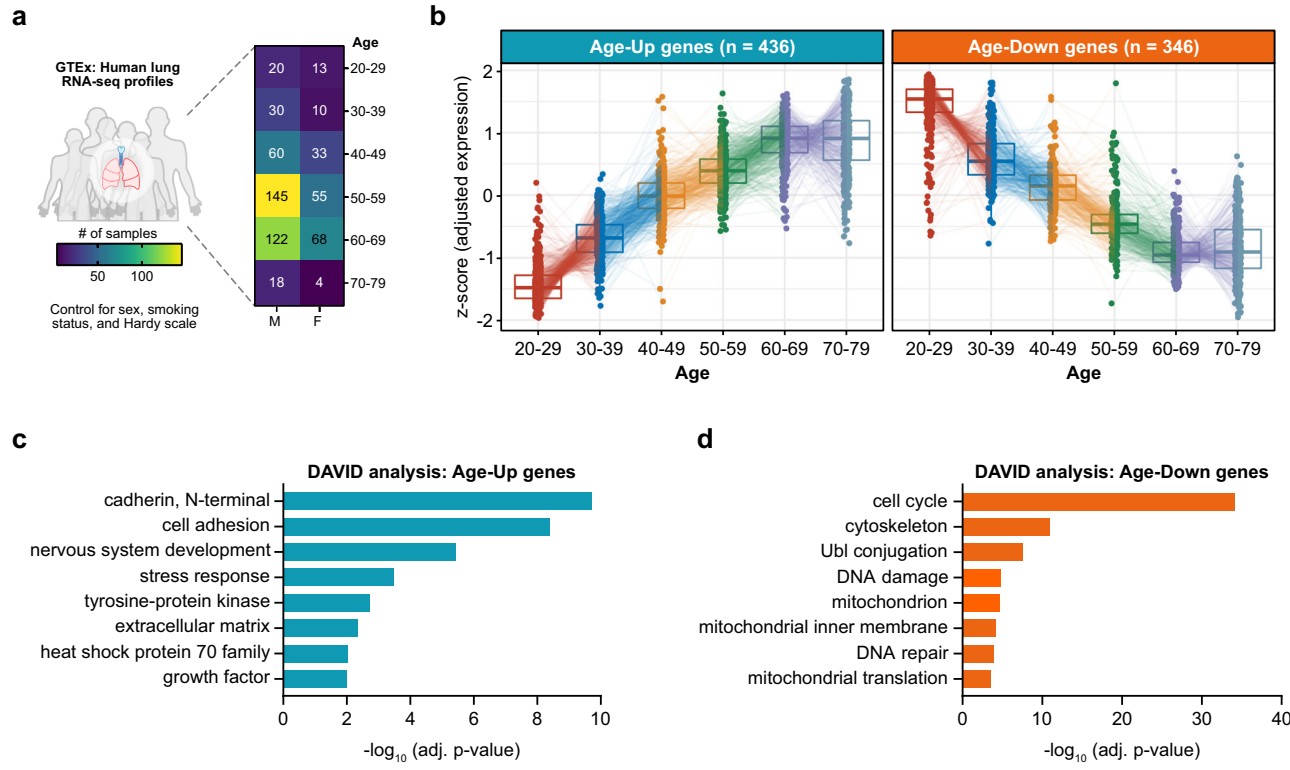

**Fig. 1 Identification of age-associated genes in the human lung. a** Demographics of the human lung RNA-seq profiles in the GTEx dataset, detailed by sex and age group ($n = 578$ samples). Downstream analyses were controlled for sex, smoking status, and Hardy scale, retaining samples with complete clinical annotations ($n = 561$). **b** Tukey boxplots (interquartile range (IQR) boxes with 1.5× IQR whiskers) of age-associated genes in the human lung. Age-Up genes increase in expression with aging (left, $n = 436$), while Age-Down genes decrease in expression with aging (right, $n = 346$). Statistical significance was assessed by DESeq2 two-sided likelihood-ratio test (adj. $p < 0.05$) controlling for sex, smoking status, and Hardy scale. Data are shown in terms of median z-score of gene expression, after adjustment for sex, smoking status, and Hardy scale. **c** DAVID gene ontology and pathway analysis of Age-Up genes. **d** DAVID gene ontology and pathway analysis of Age-Down genes.

function may therefore have unanticipated effects on COVID-19 pathogenesis[41–43].

**Cell type-specific characterization of age-associated genes.** Having compiled a set of age-associated genes, we sought to identify the lung cell types that normally express these genes, using the human lung single-cell RNA-seq (scRNA-seq) dataset from the Human Lung Cell Atlas (HLCA)[44]. By examining the scaled percentage of expressing cells within each cell subset, we identified age-associated genes predominantly enriched in different cell types (Supplementary Data 10). Cell types with the highly enriched expression for certain Age-Up genes included various fibroblast populations, muscle cells, ciliated cells, and neuroendocrine cells (Fig. 2a, Supplementary Data 11). Cell types with the highly enriched expression for certain Age-Down genes included several proliferative cell populations (Supplementary Data 12), as well as multiple epithelial cell subsets such as alveolar epithelial type 2 (AT2) cells (Fig. 2b, Supplementary Data 13). Similar results were found using an independent human lung scRNA-seq dataset from the Tissue Stability Cell Atlas (TSCA) (Supplementary Fig. 4a, b, Supplementary Data 14, Supplementary Data 15, Supplementary Data 16, Supplementary Data 17)[45].

Among the airway smooth muscle-enriched genes that were annotated as Age-Up genes, *ITIH3* and *PDGFRB* showed particularly strong age-associated increases in expression (Fig. 2c). A single nucleotide variant in *ITIH3* has been associated with heightened risk for myocardial infarctions, possibly due to its effect on increasing *ITIH3* expression[46], while platelet-derived growth factor (PDGF) signaling contributes to the development

of various age-associated lung diseases, such as pulmonary arterial hypertension and fibrosis[47,48]. Among the AT2-enriched genes that decreased in expression with age, *FOXA2* and *ORM2* were among the top-ranked genes. FOXA2 is a critical regulator of lung development[49,50] and lung tumor cell identity[51,52], whereas the function of *ORM2* in the lung is presently unknown. Thus, integrative analysis of bulk and single-cell transcriptomes revealed that many of the age-associated transcriptional changes in the human lung can be mapped to specific cell subpopulations, suggesting that the abundance of these cell types, their transcriptional status, or both, may be altered with aging.

**The cellular landscape of the aging human lung.** As the pathophysiology of viral-induced lung injury involves an intricate interplay of diverse cell types[53,54], aging-associated shifts in the lung cellular milieu[33] could contribute an important dimension to the relationship between age and risk of severe disease in patients with COVID-19[55]. To investigate the cellular landscape of the aging lung, we deconvoluted the bulk lung GTEx transcriptomes with CIBERSORTx[56], using the scRNA-seq data from the HLCA as a reference (Supplementary Fig. 5a, Supplementary Data 18). Since bulk RNA-seq measures the average expression of genes within a cell population, such datasets will reflect the relative abundances of the cell types that comprised the input population, though with the caveat that cell types can have overlapping expression profiles and such profiles may be altered in response to stimuli. To increase confidence in downstream analyses, we retained only the cell types for which their estimated proportions

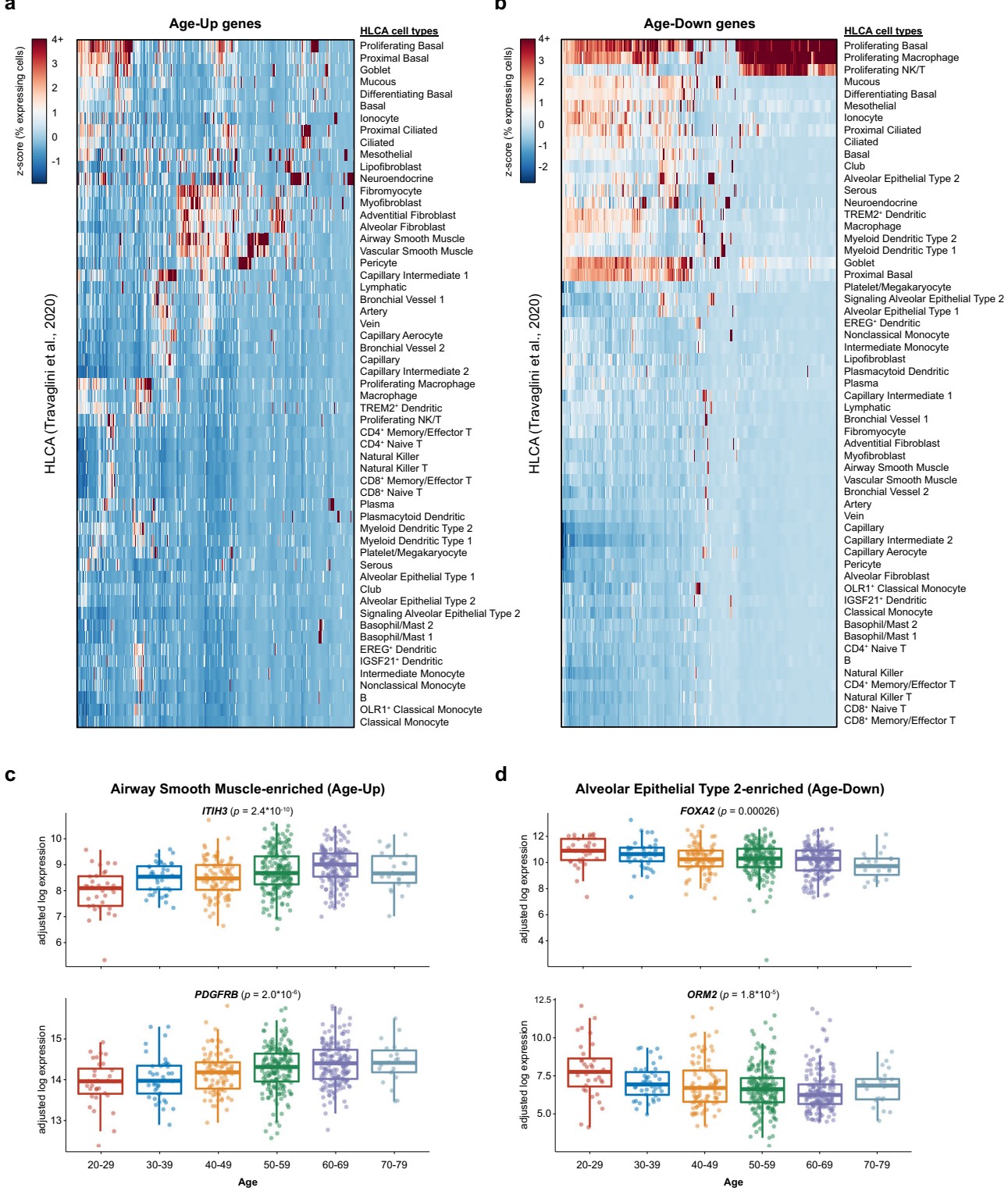

**Fig. 2 Lung single-cell transcriptomics pinpoints cell type-specific expression of age-associated genes. a** Heatmap showing the percentage of cells expressing each of the Age-Up genes (increasing with age), scaled by gene across the different cell types. Data are from the Human Lung Cell Atlas[44]. **b** Heatmap showing the percentage of cells expressing each of the Age-Down genes (decreasing with age), scaled by gene across the different cell types. Data are from the Human Lung Cell Atlas[44]. **c, d** Tukey boxplots (interquartile range (IQR) boxes with 1.5× IQR whiskers) showing the expression of *ITIH3* and *PDGFRB* (**c**), or *FOXA2* and *ORM2* (**d**), across different age groups ($n = 561$ samples). Data are shown as log-transformed expression values, adjusted for sex, smoking status, and Hardy scale. Statistical significance of age-associated variation was assessed by two-sided Kruskal–Wallis test on the adjusted expression values.

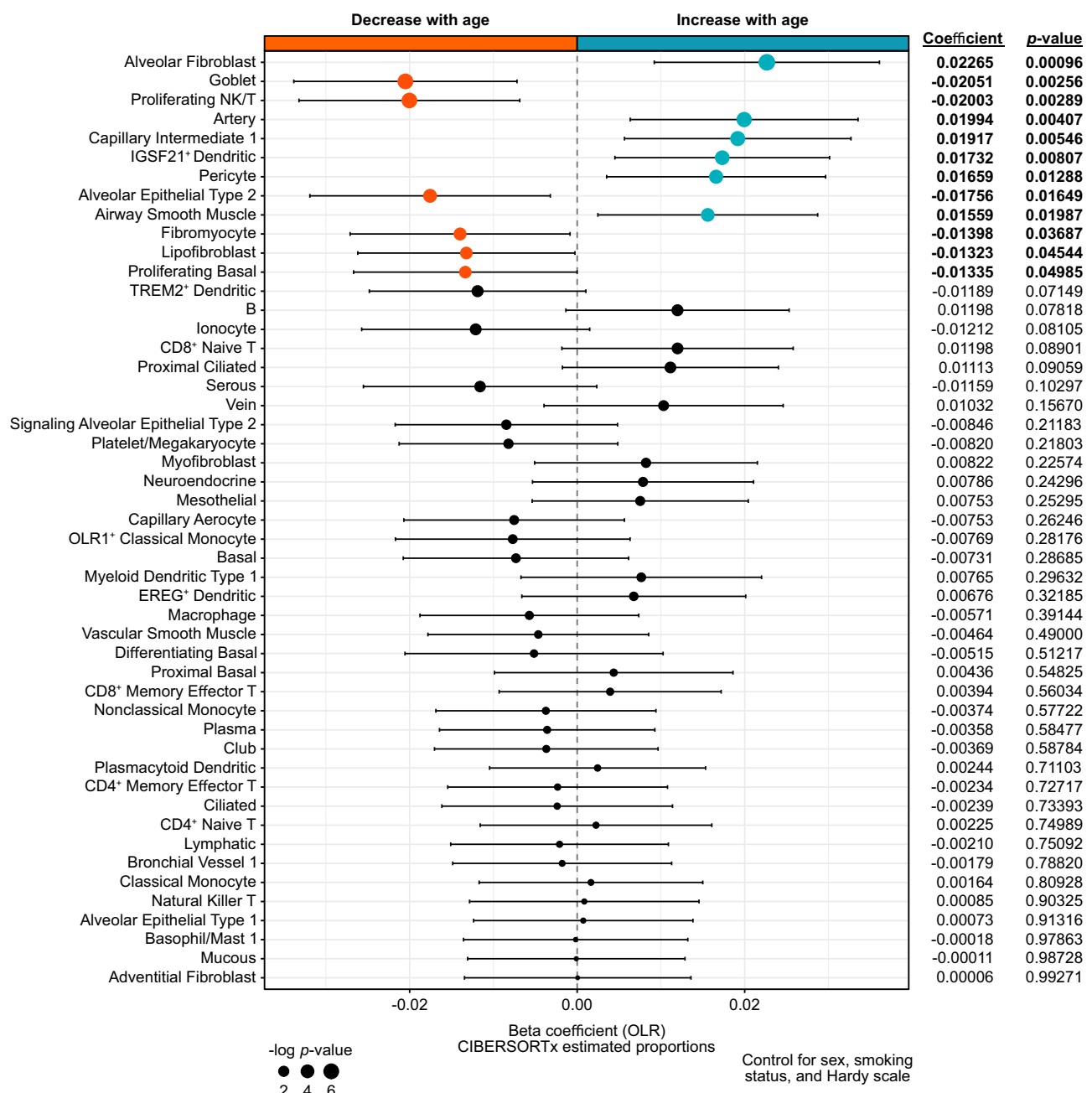

**Fig. 3 The evolving cellular landscape of the aging human lung.** Forest plot of age-associated changes in the proportions of lung cell types. Only the cell types with non-zero estimated proportions in >50% of samples were retained for analysis (49 of 57 total). Positive coefficients denote cell types that increase in proportion with age, while negative coefficients indicate cell types that decrease in proportion with age. Statistical significance of age-association was determined by a nonparametric ordinal logistic regression model, controlling for sex, smoking status, and Hardy scale. Point sizes are scaled by statistical significance. Error bars indicate 95% confidence intervals.

were >0 in at least 50% of the samples (49 out of 57 total cell types; Supplementary Fig. 5b).

Using an ordinal logistic regression model controlling for sex, smoking status, and Hardy scale, we identified age-associated alterations in the proportions of several cell types in the lung (Fig. 3, Supplementary Data 19). We observed that the proportions of multiple epithelial cell subsets (goblet cells, AT2 cells, and proliferating basal cells) significantly decreased with age (Supplementary Figs. 6 and 7). Of note, AT2 cells constitute the stem cell population of the alveoli[57,58], while basal cells function as stem/progenitor cells in the airways[59,60]. These findings are therefore consistent with the progressive loss of lung parenchyma

due to the reduced regenerative capacity of the aging lung[61]. On the other hand, the estimated proportions of alveolar fibroblasts, arterial endothelial cells, capillary intermediate endothelial cells, pericytes, and airway smooth muscle cells increased with age (Fig. 3). These data are consistent with the heightened risk for chronic obstructive pulmonary disease and pulmonary fibrosis in older populations[62], as alveolar fibroblasts[63,64], pericytes[65], and airway smooth muscle cells[66] have been directly implicated in the pathogenesis of these diseases. Collectively, these changes in the regenerative capacity and cellular architecture of the aging lung could contribute to the increased risk of COVID-19 morbidity and mortality in older patients.

Among the immune cell populations, proliferating natural killer (NK)/T cells decreased with age, whereas *IGSF21*+ dendritic cells increased with age (Fig. 3). T cells are thought to play essential roles in immune responses against SARS-CoV-2[67–70], and proliferating T cells are indicative of active antiviral responses in COVID-19 patients[71]. *IGSF21*+ dendritic cells are defined by the high expression of genes that have been previously implicated in asthma, such as *CCL2*, *CCL13*, and *IGSF21* itself[44]. Patients with severe asthma (requiring recent use of an oral corticosteroid) were found to have an increased risk of COVID-19-related death[3]. Thus, age-associated alterations in specific lung immune populations may contribute to the relationship between aging and COVID-19 severity.

**Functional roles of age-associated genes in SARS-CoV infection.** We next explored the roles of lung age-associated genes in host responses to viral infection, first searching for data on SARS-CoV. While SARS-CoV and SARS-CoV-2 belong to the same genus (*Betacoronaviridae*) and are conserved to some extent[9], they are nevertheless two distinct viruses with different epidemiological features, indicating unique virology and host biology. Therefore, data from experiments performed with SARS-CoV must be interpreted with caution. We reassessed the results from a prior in vitro siRNA screen of host factors involved in SARS-CoV infection[72]. In this kinase-focused screen, 130 factors were determined to have a significant effect on SARS-CoV replication. Seven of the 130 factors exhibited age-associated gene expression patterns (Supplementary Fig. 8a, b), with 3 genes in the Age-Up group and 4 genes in the Age-Down group (Supplementary Data 20). Using the human lung scRNA-seq data, we further determined which cell types predominantly express these host factors (Supplementary Fig. 8c, d), revealing cell type-enriched expression patterns for several of these genes.

**Age-associated host factors that interact with SARS-CoV-2 proteins.** We then investigated whether proteins encoded by age-associated genes in the human lung interact with SARS-CoV-2 proteins. A recent study interrogated the human host factors that interact with 27 different SARS-CoV-2 proteins[73], revealing the SARS-CoV-2: human protein interactome in cell lines expressing recombinant SARS-CoV-2 proteins. By cross-referencing the interacting host factors with the set of age-associated genes, we identified 14 factors at the intersection (Fig. 4a, Supplementary Data 21). Seven of these genes showed an increase in expression with age (i.e., Age-Up genes), while 7 decreased in expression with age (Age-Down genes). Mapping these factors to their interacting SARS-CoV-2 proteins, we noted that the age-associated host factors which interact with M, Nsp5, Nsp8, Nsp13, and Orf9b mostly decreased in expression with aging (Fig. 4b). In contrast, the host factors that interact with Nsp9, Nsp12, and Orf8 mostly increased in expression with age (Fig. 4c). Nsp12 encodes for the primary RNA-dependent RNA polymerase (RdRp) of SARS-CoV-2 and is a prime target for developing therapies against COVID-19[74,75]. Orf8 has been suggested to promote immune evasion by downregulating antigen presentation in SARS-CoV-2-infected cells[76]. Age-associated changes in these various host factors may thereby influence the capacity for SARS-CoV-2 replication and/or immune evasion.

To assess the cell type-specific expression patterns of these various factors, we further analyzed the lung scRNA-seq data from the HLCA. Of the SARS-CoV-2-interacting genes that increase in expression with age, *MYCBP2* was frequently expressed across several populations, particularly proliferating basal cells (Fig. 4d). *MYCBP2* was also expressed in 21.95% of AT2 cells. *CEP68* was preferentially expressed in basal cells, while

*MFGE8* was widely expressed across several stromal and smooth muscle populations. *FBLN5* was most frequently expressed in adventitial fibroblasts and alveolar fibroblasts. Of the SARS-CoV-2-interacting genes that decrease in expression with age, *ATP1B*, *DCTPP1*, and *HDAC2* were broadly expressed in many cell types, including AT2 cells (Fig. 4e). Analysis of the independent TSCA dataset revealed similar conclusions (Supplementary Fig. 9). Together, these analyses highlight specific age-associated factors that interact with the SARS-CoV-2 proteome, in the context of the lung cell types in which these factors are normally expressed.

**Age-associated genes are dysregulated by SARS-CoV-2 infection.** We next assessed whether SARS-CoV-2 infection directly alters the expression of lung age-associated genes. A recent study profiled the in vitro transcriptional changes associated with SARS-CoV-2 infection in different human lung cell lines[77]. We specifically focused on the data from A549 lung cancer cells, A549 cells transduced with an *ACE2* expression vector (A549-ACE2), and Calu-3 lung cancer cells. Several age-associated genes were found to be differentially expressed upon SARS-CoV-2 infection (Fig. 5a–c, Supplementary Data 22, Supplementary Data 23, Supplementary Data 24). Of note, the overlap between lung age-associated genes and SARS-CoV-2-regulated genes was statistically significant in the A549 and Calu-3 cells, though not in A549-ACE2 cells (Fig. 5d–f), suggesting a certain degree of similarity between the transcriptional changes associated with aging and with SARS-CoV-2 infection. Among the age-associated genes that were induced by SARS-CoV-2 infection in A549 and Calu-3 cells, the majority of these genes increased in expression with age (Fig. 5g–i, Supplementary Data 25). Conversely, among the age-associated genes that were repressed by SARS-CoV-2 infection, most of these genes decreased in expression with age. The directionality of SARS-CoV-2 regulation (induced or repressed) and the directionality of age-association (increase or decrease with age) was significantly associated for A549 and Calu-3 cells. However, these observations were not recapitulated in the A549-ACE2 cells. The discordant findings with A549-ACE2 cells compared to parental A549 or Calu-3 cells are unclear, but could potentially reflect the consequences of ectopic expression of ACE2 in the context of SARS-CoV-2 infection.

To identify a consensus set of age-associated genes that are regulated by SARS-CoV-2 infection in vitro, we integrated the analyses from all three cell lines. Totally, 603 genes were consistently induced by SARS-CoV-2 infection (Supplementary Fig. 10a, Supplementary Data 26). Of these, 8 genes increased in expression with age (Age-Up), and 2 genes decreased with age (Age-Down). The 2 induced genes in the Age-Down group were *CCL20* and *CXCL1*, which encode immune cell-recruiting cytokines. On the other hand, 641 genes were concordantly repressed by SARS-CoV-2 infection (Supplementary Fig. 10b), with 6 genes in the Age-Up category and 16 genes in the Age-Down category. Among the latter group, nearly half of these genes encode proteins that localize to the mitochondria (*AIFM1*, *C1QBP*, *DCTPP1*, *ENO1*, *MTCH2*, *NDUFA7*, and *TYMS*), highlighting certain commonalities in the mitochondrial changes observed with aging and with SARS-CoV-2 infection.

Within the consensus set of all 32 age-associated genes that are perturbed by SARS-CoV-2 infection, the directionality of SARS-CoV-2 regulation (induced or repressed) and the directionality of age-association (increase or decrease with age) were significantly associated (Supplementary Fig. 10c). Analysis of the human lung scRNA-seq datasets revealed the cell types that normally express these different genes (Supplementary Fig. 11a, b). In particular, the majority of the Age-Down genes repressed by SARS-CoV-2 infection were expressed across multiple epithelial cell types,

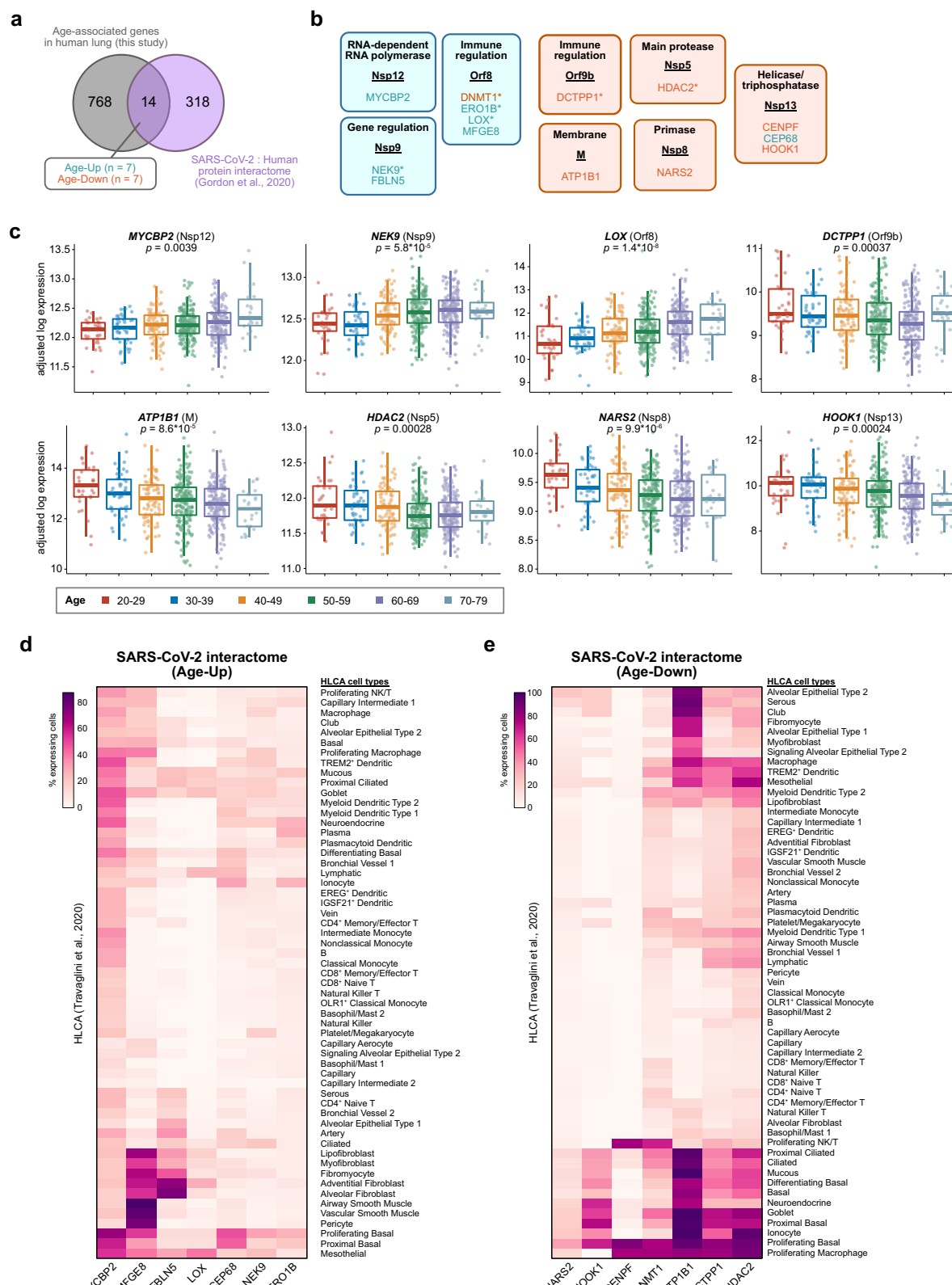

including AT2 cells (*AHCY*, *MPDU1*, *DCTPP1*, *AKR1A1*, *PARP1*, *MTCH2*, *CACYBP*, *ENO1*, *C1QBP*, *NDUFA7*, and *PHB*). Collectively, these analyses highlight the unexpected parallels between the aging transcriptome of the human lung and the transcriptional changes caused by SARS-CoV-2 infection in vitro.

We wondered whether lung age-associated genes are similarly dysregulated in patients with COVID-19. A recent study

profiled the transcriptomes of single bronchoalveolar lavage fluid cells from patients with COVID-19[78]. We collapsed the single-cell transcriptomes into pseudo-bulk transcriptomes by aggregating the data from each donor, then used these data to perform differential expression analysis. Comparing patients with severe COVID-19 to healthy controls, 6343 genes were upregulated and 3847 genes were downregulated (Fig. 6a,

**Fig. 4 Age-associated genes in the human lung interact with SARS-CoV-2 proteins. a** Venn diagram of the intersection between age-associated genes in human lung and the SARS-CoV-2: human protein interactome[73]. Of the 14 age-associated genes that were found to also interact with SARS-CoV-2, 7 of them increased in expression with age, while 7 decreased with age. **b** Age-associated genes in human lung and their interaction with SARS-CoV-2 proteins, where each block contains a SARS-CoV-2 protein (underlined) and its interacting age-associated factors. Blocks are colored by the dominant directionality of the age association (orange, decreasing with age; blue, increasing with age). Gene targets with already approved drugs, investigational new drugs, or preclinical molecules are additionally denoted with an asterisk. **c** Tukey boxplots (interquartile range (IQR) boxes with 1.5× IQR whiskers) detailing the expression of select age-associated genes that interact with SARS-CoV-2 proteins, highlighted in (**b**) ($n = 561$ samples). Data are shown as log-transformed expression values, adjusted for sex, smoking status, and Hardy scale. The SARS-CoV-2-interacting protein is annotated in parentheses. Statistical significance of the expression variation across all age groups was assessed by a two-sided Kruskal–Wallis test on the adjusted expression values. **d** Heatmap showing the percentage of cells expressing each of the Age-Up genes (increasing with age) that interact with SARS-CoV-2 proteins, as highlighted in (**b**). Data are from the Human Lung Cell Atlas[44]. **e** Heatmap showing the percentage of cells expressing each of the Age-Down genes (decreasing with age) that interact with SARS-CoV-2 proteins, as highlighted in (**b**). Data are from the Human Lung Cell Atlas[44].

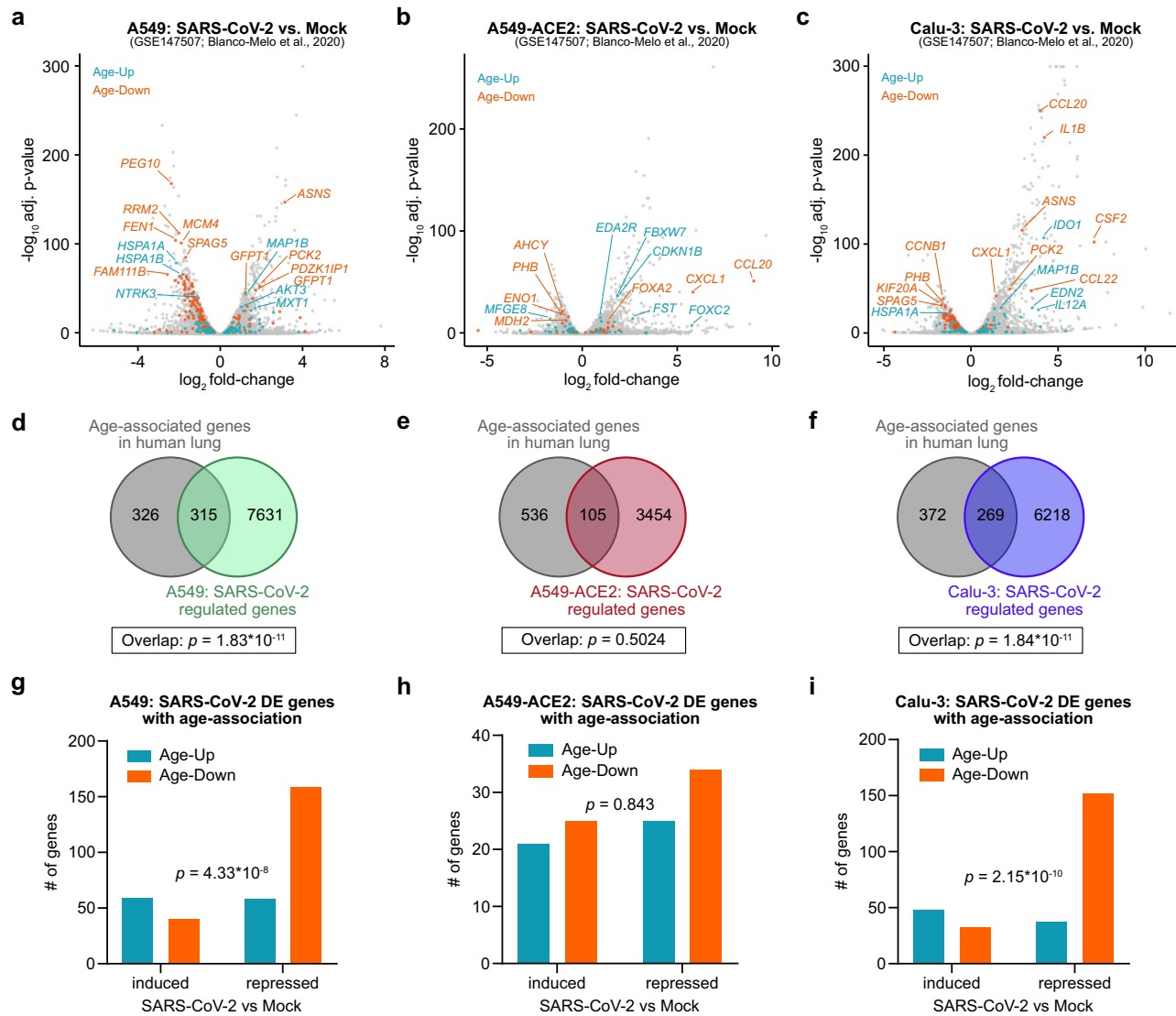

**Fig. 5 SARS-CoV-2 infection alters the expression of lung age-associated genes in vitro. a–c** Volcano plots of differentially expressed genes in A549 cells (**a**), A549 cells transduced with an *ACE2* vector (A549-ACE2) (**b**), or Calu-3 cells (**c**). Data are from GSE147507[77]. Age-associated genes are color-coded. **d–f** Venn diagrams highlighting the intersections between lung age-associated genes and SARS-CoV-2-regulated genes in A549 cells (**d**), A549-ACE2 cells (**e**), or Calu-3 cells (**f**). Statistical significance of the overlap was assessed by the hypergeometric test. **g–i** Characteristics of age-associated genes that are affected by SARS-CoV-2 infection in A549 cells (**g**), A549-ACE2 cells (**h**), or Calu-3 cells (**i**), from **d–f**. Statistical significance of the interaction between the directionality of SARS-CoV-2 regulation (induced or repressed) and the directionality of age-association (increase or decrease with age) was assessed by two-tailed Fisher's exact test.

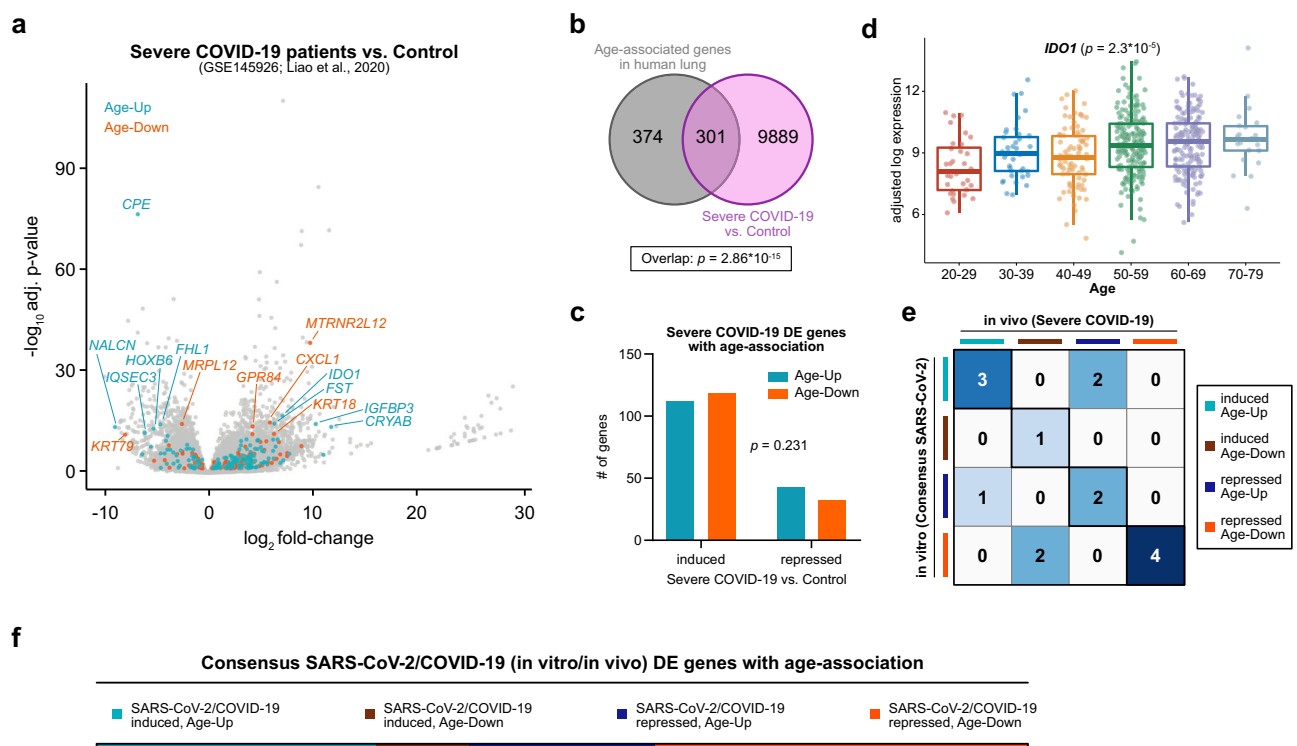

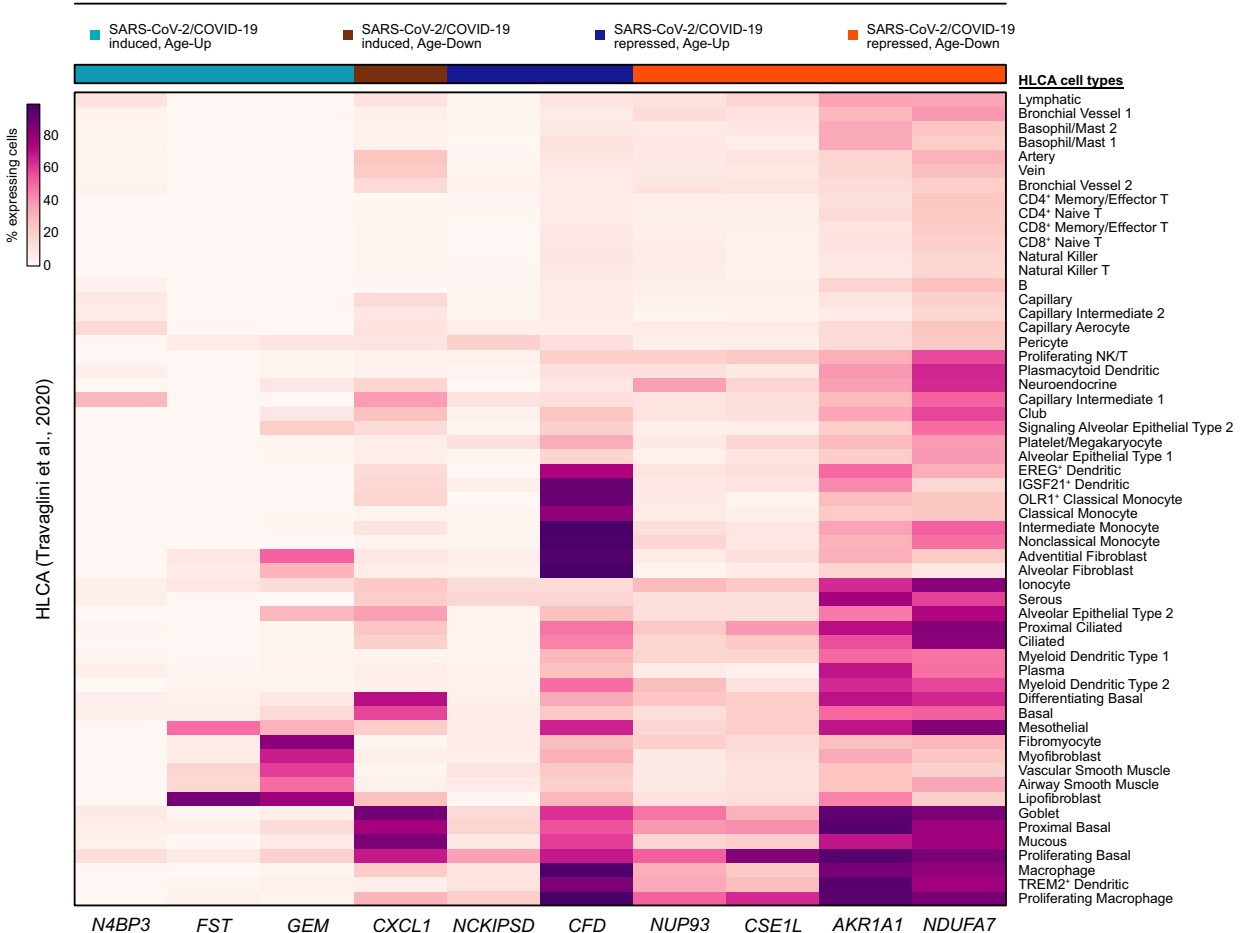

Consensus SARS-CoV-2/COVID-19 (in vitro/in vivo) DE genes with age-association

Supplementary Data 27). We again observed that age-associated genes were enriched among the genes differentially expressed in severe COVID-19 patients (Fig. 6b). However, unlike the analysis of SARS-CoV-2 infection in vitro, the directionality of COVID-19 association (induced or repressed) and the directionality of age-association (increase or decrease with age) was not significantly associated (Fig. 6c, Supplementary Data 28). Among the Age-Up genes that were upregulated in severe COVID-19 patients, *IDO1* was the most significant (Fig. 6d). *IDO1* encodes indoleamine-2,3-dioxygenase, a metabolic enzyme that has been demonstrated to have antiviral functions in a variety of contexts[79,80].

**Fig. 6 Transcriptional parallels between severe COVID-19 and the aging human lung. a** Volcano plot of differentially expressed genes in bronchoalveolar lavage fluid samples from patients with severe COVID-19 compared to healthy controls. Data are from GSE145926[78]. Age-associated genes are color-coded. **b** Venn diagram of the intersection between lung age-associated genes and differentially expressed genes in patients with severe COVID-19. Statistical significance of the overlap was assessed by the hypergeometric test. **c** Characteristics of age-associated genes that are dysregulated in patients with severe COVID-19. Statistical significance of the interaction between the directionality of COVID-19 regulation (induced or repressed) and the directionality of age-association (increase or decrease with age) was assessed by two-tailed Fisher's exact test. **d** Tukey boxplots (interquartile range (IQR) boxes with 1.5× IQR whiskers) detailing the lung expression of *IDO1* across age groups ($n = 561$ samples). Data are shown as log-transformed expression values, adjusted for sex, smoking status, and Hardy scale. Statistical significance of the expression variation across all age groups was assessed by a two-sided Kruskal–Wallis test on the adjusted expression values. **e** Heatmap detailing the categories of the age-associated genes that are differentially expressed upon SARS-CoV-2 infection in vitro and in patients with severe COVID-19. **f** Heatmap showing the percentage of cells expressing the consensus SARS-CoV-2/COVID-19-regulated genes that also exhibit age-associated expression, from **e**. Genes are annotated by whether they are induced or repressed by SARS-CoV-2/COVID-19, and whether they increase (Age-Up) or decrease (Age-Down) in expression with age. Data are from the Human Lung Cell Atlas[44].

We subsequently intersected the age-associated genes that are differentially expressed upon SARS-CoV-2 infection in vitro or in severe COVID-19 patients (Fig. 6e, Supplementary Data 29). A total of ten age-associated genes showed consistent patterns of differential expression both in vitro and in vivo. Turning to the lung scRNA-seq data, we examined which lung cell types normally express these genes (Fig. 6f and Supplementary Fig. 12). Whereas *AKR1A1* and *NDUFA7* were widely expressed in multiple cell populations, *CFD* was preferentially expressed in innate immune cells, while *FST* and *GEM* were primarily expressed in stromal and muscle cells. Interestingly, *CXCL1* was most highly expressed in airway epithelial populations (basal cells, goblet cells, mucous cells). CXCL1 is a potent neutrophil attractant[81,82], and neutrophil-derived extracellular traps have been frequently observed in patients that died from COVID-19[83]. Taken together, these data demonstrate that age-associated genes are frequently dysregulated in patients with severe COVID-19, further identifying the specific cell types in the lung that normally express these genes.

## Discussion

Here, we systematically analyzed the transcriptome and cellular landscape of the aging human lung in relation to SARS-CoV-2. We found that the aging lung is characterized by a wide array of changes that could contribute to the worse outcomes of older patients with COVID-19. Controlling for sex, smoking status, and Hardy scale, we identified 782 genes that exhibit age-associated expression patterns. We subsequently demonstrated that the aging lung is characterized by several gene signatures, including increased cell adhesion, heightened stress responses, reduced mitochondrial activity, and decreased proliferation. By integrating these data with single-cell transcriptomes of human lung tissue, we further pinpointed the specific cell types that normally express the age-associated genes. Through bulk deconvolution analysis, we showed that the proportions of lung stem/progenitor populations (namely AT2 cells and proliferating basal cells) decrease with age, whereas the proportions of alveolar fibroblasts, pericytes, endothelial cells, and airway smooth muscle cells increase with age. Among the immune cell types in the lung, proliferating NK/T cells decrease in proportion with age, while *IGSF21*[+] dendritic cells increase with aging.

We also found that some of the age-associated factors have been shown to directly interact with the SARS-CoV-2 proteome. Furthermore, age-associated genes are enriched among genes directly regulated by SARS-CoV-2 infection in vitro, suggesting transcriptional parallels between the aging lung and SARS-CoV-2 infection. These findings were recapitulated in vivo when comparing patients with severe COVID-19 to healthy controls. We speculate that the transcriptional parallels between aging and SARS-CoV-2 infection/COVID-19 could reflect the dysregulation

of cellular pathways common to both processes. However, while the directionality of age-association and SARS-CoV-2 regulation was concordant in both A549 and Calu-3 cell lines, we did not observe the same patterns in A549-ACE2 cells or in patients with severe COVID-19. Exploring the commonalities and differences between aging and SARS-CoV-2 infection in these different contexts may serve as a valuable window into the mechanisms by which COVID-19 differentially affects patients across the lifespan.

We emphasize that the analyses presented here cannot be used to guide clinical practice at this stage. Whether any of these age-associated changes causally contribute to the heightened susceptibility of COVID-19 in older populations remains to be experimentally tested. Our analyses unveil a number of observations and phenomena that provide potential directions for subsequent research efforts on SARS-CoV-2, generating genetically tractable hypotheses for why advanced age is one of the strongest risk factors for COVID-19 morbidity and mortality.

## Methods

**Data accession**. The Genotype-Tissue Expression (GTEx) project was supported by the Common Fund of the Office of the Director of the National Institutes of Health, and by NCI, NHGRI, NHLBI, NIDA, NIMH, and NINDS[13,14]. RNA-seq raw counts and normalized TPM matrices were directly downloaded from the GTEx Portal (https://gtexportal.org/home/datasets) on March 18, 2020, release v8. Gene expression data used in this study are publicly available on the web portal and have been de-identified, except for patient age range and gender. Detailed clinical annotations of the GTEx cohort were obtained as controlled access data through dbGaP (phs000424.v8.p2).

Single-cell transcriptomes of human lungs were directly obtained from the HLCA (https://github.com/krasnowlab/HLCA) (Synapse #syn21041850)[44], and from the Tissue Stability Cell Atlas (https://www.tissuestabilitycellatlas.org/)[45] (PRJEB31843). In both cases, preprocessed R objects were downloaded from the respective repositories and utilized for downstream analysis.

Transcriptomic profiles of cell lines infected with SARS-CoV-2 in vitro were obtained from the Gene Expression Omnibus (GSE147507). Transcriptomic profiles of patients with severe COVID-19 and healthy controls were also obtained from the Gene Expression Omnibus (GSE145926).

**Analysis of clinical features associated with expression of SARS-CoV-2 entry factors**. Clinical annotations for age, sex, obesity, hypertension, type 1 diabetes, type 2 diabetes, Hardy scale, and smoking history were compiled from the controlled access GTEx metadata. A Hardy scale of 1 was used as the reference for comparisons (Hardy scale 0: on ventilator prior to death, 1: violent and fast death, 2: fast death of natural causes, 3: intermediate death, 4: slow death from chronic illness). With these annotations, a multivariable linear regression model was utilized to assess whether different clinical features were associated with the log-transformed expression of SARS-CoV-2 entry factors. The resulting estimated regression coefficients were visualized as forest plots with 95% confidence intervals.

**Identification of age-associated genes in human lung**. To identify age-associated genes, the RNA-seq raw count matrix was analyzed by DESeq2 (v1.24.0)[32], using the likelihood-ratio test (LRT). Sex, smoking status, and Hardy scale were included in the LRT model to control for these factors. Donor ages were binned into decades (e.g., 20–29, 30–39, 40–49, 50–59, 60–69, 70–79) for analysis. Age-associated genes were determined at a significance threshold of adjusted $p < 0.05$. Genes

passing the significance threshold were then scaled to z-scores and clustered using the degPatterns function from the R package DEGreport (v1.20.0). Gene clusters with progressive and consistent trends with age were retained for downstream analysis.

Gene ontology and pathway enrichment analysis were performed using DAVID (v6.8)[84] (https://david.ncifcrf.gov/), separating the age-associated genes into two primary clusters (increasing or decreasing with age).

**RNA-seq gene expression visualization and statistical analysis.** For visualization of unadjusted RNA-seq expression data, the TPM values were $\log_2$ transformed and plotted in R (v3.6.1). For visualization of adjusted expression levels, the raw counts were first processed by variance-stabilizing transformation with DESeq2[32], followed by statistical adjustment for sex, smoking status, and Hardy scale using the remove Batch Effect function in limma[85] (v3.45). All boxplots are Tukey boxplots, with interquartile range (IQR) boxes and 1.5× IQR whiskers. Pairwise statistical comparisons in the plots are assessed by the two-tailed Mann–Whitney test, while statistical comparisons across all age groups were performed by Kruskal–Wallis test. We note that the identification of age-associated genes was purely determined through the two-sided DESeq2 LRT[32] described above; the two-sided Mann–Whitney or Kruskal–Wallis statistics shown on the plots are solely for confirmatory purposes.

**Normal human lung scRNA-seq data analysis.** scRNA-seq data were analyzed in R (v3.6.1) using Seurat[86,87] (v3.2) and custom scripts. Of the 782 age-associated genes identified from GTEx bulk transcriptomes, 712 genes were matched in the Tissue Stability Cell Atlas dataset and 683 genes were matched in the HLCA dataset. To determine the percentage of cells expressing a given gene, the expression matrices were converted to binary matrices by setting a threshold of expression >0. Cell type-specific expression frequencies for each gene were then calculated using the provided cell type annotations. To identify genes preferentially expressed in a specific cell type, we further scaled the expression frequencies in R to obtain z-scores. Data were visualized in R using the NMF package[88] (v0.23).

**Inferring the cellular composition of the aging human lung.** To infer the cellular composition of each bulk lung transcriptome, we used the CIBERSORTx algorithm[56]. We provided the HLCA dataset[44] as a reference to calculate estimated cell-type proportions in each lung sample with S-mode batch correction. The resultant cell type proportions were analyzed in R. Cell types with estimated proportions of "0" in >50% of samples were filtered out prior to further analysis. Statistical significance of age-association was assessed by an ordinal logistic regression model, a generalization of the nonparametric Kruskal–Wallis test that allows for multifactorial designs. Sex, smoking status, and Hardy scale were included in the regression model to pinpoint the cell types that are specifically altered with aging. The estimated coefficients were visualized as a forest plot with 95% confidence intervals. For visualization purposes, the estimated proportions were adjusted for sex, smoking status, and Hardy scale by extracting the residuals from fitting a generalized linear model with these variables.

**Assessing the functional roles of lung age-associated genes in SARS-CoV.** To assess whether any age-associated genes affect host responses to SARS-CoV (a coronavirus related to SARS-CoV-2), we analyzed the data from a published siRNA screen of host factors influencing SARS-CoV[72] (Data Set S1 in the publication; accessed on March 20, 2020). For data visualization, each point corresponding to a target gene was size-scaled and color-coded according to the age-association statistical analyses described above.

**Age-associated genes that interact with the SARS-CoV-2 proteome.** To assess whether any lung age-associated genes encode proteins that interact with the SARS-CoV-2 proteome, we compiled the data from a study detailing the human host factors that interact with 27 different proteins in the SARS-CoV-2 proteome[73] (corresponding preprint manuscript accessed on March 23, 2020).

**Age-associated genes that are transcriptionally regulated upon SARS-CoV-2 infection in vitro.** To assess whether the expression of lung age-associated genes is influenced by SARS-CoV-2 infection, we utilized the data from a recent study detailing the transcriptional response to SARS-CoV-2 infection[77], from the Gene Expression Omnibus (GSE147507) (accessed on April 13, 2020). Differentially expressed genes were determined using the Wald test in DESeq2 (v1.24.0)[32] comparing SARS-CoV-2-infected cells to batch-matched mock controls, with a significance threshold of adjusted $p < 0.05$.

Of the 782 age-associated genes, 641 genes were matched to the RNA-seq dataset. Statistical significance of overlaps between the gene sets was assessed by two-tailed hypergeometric test, assuming 21,797 total genes as annotated in the RNA-seq dataset and 641 age-associated genes. Statistical significance of the association between the directionality of SARS-CoV-2 regulation and the directionality of age-association was assessed by two-tailed Fisher's exact test.

**Age-associated genes that are transcriptionally dysregulated in patients with severe COVID-19.** To assess whether the expression of lung age-associated genes is affected in patients with severe COVID-19, we utilized the data from a recent study detailing the transcriptomes of bronchioalveolar lavage fluid cells from patients with COVID-19[78], from the Gene Expression Omnibus (GSE145926) (accessed on May 14, 2020). Cells were filtered in a similar manner as previously described by the study authors (unique RNA species ≥200 and ≤6000, mitochondrial reads ≤10%, and UMI ≥ 1000). The single-cell transcriptomes were collapsed into pseudo-bulk profiles by summing the read counts for all the cells from each donor. Differentially expressed genes were then determined using the Wald test in DESeq2 (v1.24.0)[32] comparing patients with severe COVID-19 to healthy controls, with a significance threshold of adjusted $p < 0.05$.

Of the 782 age-associated genes, 675 genes were matched to the scRNA-seq pseudo-bulk dataset. Statistical significance of overlaps between the gene sets was assessed by hypergeometric test, assuming 33,540 total genes as annotated in the scRNA-seq dataset and 675 age-associated genes. Statistical significance of the association between the directionality of COVID-19 regulation and the directionality of age-association was assessed by two-tailed Fisher's exact test.

**Statistical information summary.** Comprehensive information on the statistical analyses used is included in various places, including the figures, figure legends, and results, where the methods, significance, p values, and/or tails are described. All error bars have been defined in figure legends or methods.

**Reporting summary.** Further information on research design is available in the Nature Research Reporting Summary linked to this article.

## Data availability

All relevant processed data generated during this study are included in this article and its supplementary information files. Raw data are from various sources as described above. Accession codes: GTEx (phs000424.v8.p2), Human Lung Cell Atlas (#syn21041850), Tissue Stability Cell Atlas (PRJEB31843), SARS-CoV-2 infection in vitro (GSE147507), and COVID-19 patients (GSE145926). All data related to this study are freely available from the links provided in the Data Accession section of the Methods or from the corresponding author upon request, with the exception of detailed clinical annotations on the GTEx cohort that are under controlled access.

## Code availability

Codes used for data analysis or generation of the figures related to this study have been deposited to GitHub (https://github.com/rdchow/agingLung-COVID)[89].

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

## Acknowledgements

We thank Akiko Iwasaki, Craig Wilen, Hongyu Zhao, Wei Liu, Wenxuan Deng, Andre Levchenko, Katie Zhu, Ruth Montgomery, Bram Gerritsen, Steven Kleinstein, Richard Sutton, Dan DiMaio, Yong Xiong, Brett Lindenbach, Albert Ko, and a number of other colleagues for discussions, where critical comments and suggestions were incorporated into the analyses and manuscript. We thank Antonio Giraldez, Andre Levchenko, Chris Incarvito, Mike Crair, and Scott Strobel for their support on COVID-19 research. We thank our colleagues in the Chen Lab, the Genetics Department, the Systems Biology Institute, and various Yale entities. We thank the High-Performance Computing Center at Yale for technical support. R.D.C. is supported by the NIH Medical Scientist Training Program Training Grant (T32GM007205) and NIH/NCI (F30CA250249). S.C. is supported by NIH/NCI/NIDA (DP2CA238295, 1R01CA231112, U54CA209992-8697, R33CA225498, and RF1DA048811) and Do.D. (W81XWH-20-1-0072 and PR201784). This work is supported by the Chen Lab discretionary funds.

## Author contributions

R.C. and S.C. conceived and designed the study. R.C. developed the analysis approach, performed data analyses, and created the figures. M.M. assisted with the data analyses. R.C. and S.C. prepared the paper. S.C. supervised the work.

## Competing interests

The authors have no competing interests as defined by Nature Research, or other interests that might be perceived to influence the interpretation of the article. As a note for full disclosure, S.C. is a co-founder, funding recipient and scientific advisor of EvolveImmune Therapeutics, which is not related to this study.
