## [Peer Review File · Nature Communications]

REVIEWER COMMENTS

Reviewer #1 (Remarks to the Author):

Statistics across multiple countries show that COVID19 severity and fatality rate increases exponentially with age, but the explanation for such dependence is not clear. A few hypotheses (A. Wang et al. 2020; Santesmases et al. 2020) have been recently proposed: for example, ageing of the immune system prevents an effective response to a new infection, increased rate of comorbidities in elderly (such as hypertension and CVD) as well as change in the expression of the key entry points for the SARS-CoV2 virus - receptor ACE2.

Chow and Chen are trying to understand this age-dependence for COVID19 by performing a comprehensive investigation of ageing lung transcriptome using publicly available bulk and single-cell RNAseq data and trying to interpret it in the light of available knowledge about SARS-CoV infection factors, SARS-CoV2 interactome and response genes.

In the first part of the paper, authors investigate change in general lung transcriptome, cell type composition with age as well as focus on change in the expression of SARS-CoV2 entry factors (ACE2, TMPRSS2 and CTSL) with age. It is worth noting, that GTEx bulk RNAseq data has been previously used to study ageing changes across multiple organs, including lung (Yang et al. 2015). However, the data had not been interpreted in light of COVID19 pathology and change of cell type composition in the lung with age was not previously inferred, both of these constitute a novelty here. The authors present correlations between age-related changes in gene expression and cell type composition and COVID19 severity in elderly, which is useful for the scientific community - however, it worth noting that there is no causality between the two has been established.

In the second part of the paper, the authors turn to available knowledge about host genes involved in SARS-CoV & SARS-CoV2 infections to investigate if any disease-relevant genes show age-related change in gene expression and try to reveal which cell types are responsible for their expression in the lung. The authors identified several genes that are known to interact with SARS-CoV2 upon entry and also change in expression with age as well as found parallels between genes expressed in response to SARS-CoV2 infection and ageing. While these are interesting observations in itself - it would be useful to get deeper insights into each of these statements. For example, how crucial are those genes that change with age and also interact with SARS-CoV2 for viral infection? How do authors explain parallels between gene expression change with age and upon SARS-CoV2 infection (are they part of general stress, for example)?

To conclude, we think the authors present a very interesting and relevant analysis of ageing changes in the lung in the light of available knowledge about SARS-CoV2. The authors presented evidence for several different mechanisms that can contribute to the high severity of COVID19 in elderly (with no causation established), which can be followed up in the future. However, we want to note some concerns about the authors' approach for finding ageing-related genes and performing cell type composition analysis (which are more detailed in major points). Finally, improved structuring and better headings would improve the flow of the manuscript (see minor points for more detail).

Major points:

1. The authors claim that SARS-CoV2 entry factors (ACE2, TMPRSS2 and CTSL) do not change with age in bulk data and thus say 'entry factors alone is unlikely to explain the relationship between age and severity of COVID-19 illness'. Although it is possible that expression of entry factors alone are unlikely to fully explain the relationship between age and COVID19 illness - we consider that the absence of this relationship in bulk data specifically does not prove or disprove this claim. Gene expression in specific cell types may go up or down with age, leading to no overall change in bulk data, but likely functionally relevant changes at single cell level. In fact, a few preprints document such changes for SARS-CoV2 entry factors in specific lung cell types with age (A. Wang et al. 2020; Muus et al., n.d.).

2. In the analysis of GTEx data, the authors do not account for the effect of covariates such as sex, smoking status (if available) as well as patient being on a ventilator before death, which can be essential to correctly discern the effect of age on the expression of SARS-CoV2 entry factors as well as other genes. For example, stratifying patients on ventilator and not on ventilator is important, as it has been shown before that ACE2 expression may be crucial in recovering from ventilator injuries (D. Wang et al. 2019). Accounting for this metadata may lead to a different dependence between ACE2 expression in bulk lung data as another preprint shows (Santesmasses et al. 2020).

3. The authors use a 'gene signature based approach' to identify cell type composition, which relies on the publicly available bulk RNAseq datasets obtained from cell lines or FACS sorted cell types not necessarily from the lung. Because of that we advise using a single-cell reference obtained from the lung to perform the cell type decomposition in the lung, which may lead to more accurate results. If that is not possible - it is at least worth mentioning the possible disadvantages of using 'signature based' as opposed to deconvolution approach in the discussion.

4. When describing change in T cell composition with age in lung the authors should cite extensive research work done on that topic and findings across various tissues, including lung (Thome et al. 2014; Kumar, Connors, and Farber 2018).

Minor points:

1. Page 4, mention number of donors for lung samples in GTEx;
2. We advise that authors separate their findings into sections, and give them different subtitles so it is easier to comprehend the main findings, for example: (i) age-related changes in lung gene expression and cell type composition; (ii) cross-reference of genes changing with age in lung with the a) genes important for infection with SARS-CoV; b) genes responding and interacting with SARS-CoV2 genes.
3. Fig.2. Mention the number of genes in Muscle-enriched Cluster 1 genes and AT2-enriched Cluster 2 genes that was used to perform gene ontology analysis;
4. "On the transcriptional level, we first identified 1,285 genes that exhibit age-associated expression patterns." - We suggest to delete "first" to avoid the inference that the authors were the first to study ageing in the GTEx data set, This analysis has been performed in the past with the aim to derive ageing gene expression signatures across multiple tissues, including lung (see, for example (Yang et al. 2015)). (The "first" may of course simply refer to the order in which the analysis was done.)
5. The authors also use different colour schemes to show the same data. A number of figures include heatmaps showing "% of expressing cells". While the scales will vary with the data, the use of completely different colour schemes is confusing.
6. Throughout the paper, the authors use two different single-cell lung datasets (Tissue stability and Lung Cell Atlas) and infer cell enrichment using a collection of publicly available datasets within XCell package, which leads to different cell type annotation on different figures and initial confusion in understanding. Thus we advise authors clearly state the origin of cell annotations in the legend (where it is not present) and in the figure itself and emphasize it in the text (for cell type enrichment analysis with XCell).

References

Kumar, Brahma V., Thomas J. Connors, and Donna L. Farber. 2018. "Human T Cell Development, Localization, and Function throughout Life." *Immunity* 48 (2): 202–13.

Muus, Christoph, Malte D. Luecken, Gokcen Eraslan, Avinash Waghay, Graham Heimberg, Lisa Sikkema, Yoshihiko Kobayashi, et al. n.d. "Integrated Analyses of Single-Cell Atlases Reveal Age, Gender, and Smoking Status Associations with Cell Type-Specific Expression of Mediators of SARS-CoV-2 Viral Entry and Highlights Inflammatory Programs in Putative Target Cells." <https://doi.org/10.1101/2020.04.19.049254>.

Santesmasses, Didac, José Pedro Castro, Aleksandr A. Zenin, Anastasia V. Shindyapina, Maxim V. Gerashchenko, Bohan Zhang, Csaba Kerepesi, Sun Hee Yim, Peter O. Fedichev, and Vadim N. Gladyshev. 2020. "COVID-19 Is an Emergent Disease of Aging." *Infectious Diseases (except HIV/AIDS)*. medRxiv. <https://doi.org/10.1101/2020.04.15.20060095>.

Thome, Joseph J. C., Naomi Yudanin, Yoshiaki Ohmura, Masaru Kubota, Boris Grinshpun, Taheri Sathaliyawala, Tomoaki Kato, Harvey Lerner, Yufeng Shen, and Donna L. Farber. 2014. "Spatial Map of Human T Cell Compartmentalization and Maintenance over Decades of Life." *Cell* 159 (4): 814–28.

Wang, Allen, Joshua Chiou, Olivier B. Poirion, Justin Buchanan, Michael J. Valdez, Jamie M. Verheyden, Xiaomeng Hou, et al. 2020. "Single Nucleus Multiomic Profiling Reveals Age-Dynamic Regulation of Host Genes Associated with SARS-CoV-2 Infection." bioRxiv. <https://doi.org/10.1101/2020.04.12.037580>.

Wang, Di, Xiao-Qing Chai, Costan G. Magnussen, Graeme R. Zosky, Shu-Hua Shu, Xin Wei, and Shan-Shan Hu. 2019. "Renin-Angiotensin-System, a Potential Pharmacological Candidate, in Acute Respiratory Distress Syndrome during Mechanical Ventilation." *Pulmonary Pharmacology & Therapeutics* 58 (October): 101833.

Yang, Jialiang, Tao Huang, Francesca Petralia, Quan Long, Bin Zhang, Carmen Argmann, Yong Zhao, et al. 2015. "Synchronized Age-Related Gene Expression Changes across Multiple Tissues in Human and the Link to Complex Diseases." *Scientific Reports* 5 (October): 15145.

Reviewed by Dr. Sarah Teichmann, Dr. Kerstin Meyer and Veronika Kedlian.

Reviewer #2 (Remarks to the Author):

This is a bioinformatics study that compares the aging transcriptome with siRNA screen of host factors involved in SARS-CoV2 infection and data from a SARS-CoV2 human protein-protein interaction map.

The results from analysis of changes in the aging lung are well done (Figs. 1-3) and reveals interesting new data. This includes the finding that the aging lung is characterized by increased vascular smooth muscle contraction, reduced mitochondrial activity, and decreased lipid metabolism. A particularly noteworthy finding is that expression of SARS-CoV2 entry factors (ACE2, TMPRSS2 and CTSL) are not significantly changed, at least at the mRNA level, supporting the notion that other host factors involving innate immunity may be more important. In this regard, the data showing changes in macro phage numbers and phenotype will be important to the field.

I am not sure the data with A549 cells and other cancer cells (Fig. 6) contribute much to our understanding, and relevance to conclusions are unclear.

Reviewer #3 (Remarks to the Author):

In this study, the authors analysed published experimental data to determine how the interaction between SARS-CoV-2 and host cell may be regulated by age-associated factors. They postulate that these age-associated genes may explain COVID-19 pathogenesis in the elderly. Unfortunately, the conclusions drawn are not supported by the data presented.

(I) Part of the analysis was performed using published data on SARS-CoV and it is not justifiable to extrapolate this to SARS-CoV-2. As the authors have also stated “they are nevertheless two distinct viruses with different epidemiological features, indicating unique virology and host biology”. Indeed, these two viruses have been shown to have different viral kinetics in human infection and also significant difference in clinical outcome. Thus, it is not clear how the experimental data on SARS-CoV strengthens this study.

(II) The experimental data used in this study do not include any study obtained from COVID-19 patients. Rather it relied on interactomic and transcriptomic studies published by other groups in human cell lines. In addition, the authors did not determine if the observations from these profiling are replicated in clinical samples taken COVID-19 patients.

(III) Most importantly, none of the age-associated genes or pathway identified in this study was functionally tested in cell culture or animal model of infection. Thus, it is not possible to know if they have any contribution to COVID-19 pathogenesis in the elderly.

Highlights of key changes for this revision:

We appreciate the insightful comments made by all the reviewers. Below are several bullet points that highlight the major changes in this revised manuscript:

- Utilized the MuSiC algorithm to deconvolute the bulk lung transcriptomes, identifying the specific lung cell types that change in abundance with aging (Reviewer #1, comment 3; **see new Figure 3**). This analysis utilizes the lung scRNA-seq dataset as a reference for cell type deconvolution, thus representing an improvement over our prior analyses using the xCell algorithm.
- Performed new multivariable regression analysis of several clinical factors and their relation to the expression of SARS-CoV-2 entry factors (Reviewer #1, comment 2; **see new Figure 7**).
- Performed new analysis of transcriptomic data from COVID-19 patients compared to healthy controls (Reviewer #2 + Reviewer #3, comment 2; **see new Figure 6**). We find that, similar to *in vitro* SARS-CoV-2 infection models, age-associated genes are heavily enriched among the differentially expressed genes in COVID-19 patients. Furthermore, the directionality of the association is conserved (COVID-19 upregulated genes tend to increase with age, and vice versa).
- Restructured and revised the manuscript text and figures to address various concerns.

Below are the point-by-point responses to each specific comment.

Point-by-point responses:

REVIEWER COMMENTS

Reviewer #1 (Remarks to the Author):

Statistics across multiple countries show that COVID19 severity and fatality rate increases exponentially with age, but the explanation for such dependence is not clear. A few hypotheses (A. Wang et al. 2020; Santessmasses et al. 2020) have been recently proposed: for example, ageing of the immune system prevents an effective response to a new infection, increased rate of comorbidities in elderly (such as hypertension and CVD) as well as change in the expression of the key entry points for the SARS-CoV2 virus - receptor ACE2.

Chow and Chen are trying to understand this age-dependence for COVID19 by performing a comprehensive investigation of ageing lung transcriptome using publicly available bulk and single-cell RNAseq data and trying to interpret it in the light of available knowledge about SARS-CoV infection factors, SARS-CoV2 interactome and response genes.

In the first part of the paper, authors investigate change in general lung transcriptome, cell type composition with age as well as focus on change in the expression of SARS-CoV2 entry factors (ACE2, TMPRSS2 and CTSL) with age. It is worth noting, that GTEx bulk RNAseq data has been previously used to study ageing changes across multiple organs, including lung (Yang et al. 2015). However, the data had not been interpreted in light of COVID19 pathology and change of cell type composition in the lung with age was not previously inferred, both of these constitute a novelty here. The authors present correlations between age-related changes in gene expression and cell type composition and COVID19 severity in elderly, which is useful for the scientific community - however, it worth noting that there is no causality between the two has been established.

In the second part of the paper, the authors turn to available knowledge about host genes involved in SARS-CoV & SARS-CoV2 infections to investigate if any disease-relevant genes show age-related change in gene expression and try to reveal which cell types are responsible for their expression in the lung. The authors identified several genes that are known to interact with SARS-CoV2 upon entry and also change in expression with age as well as found parallels between genes expressed in response to SARS-CoV2 infection and ageing. While these are interesting observations in itself - it would be useful to get deeper insights into each of these statements. For example, how crucial are those genes that change with age and also interact with SARS-CoV2 for viral infection? How do authors explain parallels between gene expression change with age and upon SARS-CoV2 infection (are they part of general stress, for example)?

To conclude, we think the authors present a very interesting and relevant analysis of ageing changes in the lung in the light of available knowledge about SARS-CoV2. The authors presented evidence for several different mechanisms that can contribute to the high severity of COVID19 in elderly (with no causation established), which can be followed up in the future. However, we want to note some concerns about the authors' approach for finding ageing-related genes and performing cell type composition analysis (which are more detailed in major points). Finally, improved

structuring and better headings would improve the flow of the manuscript (see minor points for more detail).

We thank the reviewer for the in-depth summary of this study.

Major points:

1. The authors claim that SARS-CoV2 entry factors (ACE2, TMPRSS2 and CTSL) do not change with age in bulk data and thus say ‘entry factors alone is unlikely to explain the relationship between age and severity of COVID-19 illness’. Although it is possible that expression of entry factors alone are unlikely to fully explain the relationship between age and COVID19 illness - we consider that the absence of this relationship in bulk data specifically does not prove or disprove this claim. Gene expression in specific cell types may go up or down with age, leading to no overall change in bulk data, but likely functionally relevant changes at single cell level. In fact, a few preprints document such changes for SARS-CoV2 entry factors in specific lung cell types with age (A. Wang et al. 2020; Muus et al., n.d.).

We agree with the reviewer and appreciate their nuanced perspective on this result. We have revised the manuscript to address this point that bulk transcriptional changes (or lack thereof) may not capture changes that would only be apparent at the single cell level (pg 12, lines 331-336). We also cite the manuscripts that have described age-associated increases in entry factor expression at the single-cell level (pg 12, lines 331-336).

2. In the analysis of GTEx data, the authors do not account for the effect of covariates such as sex, smoking status (if available) as well as patient being on a ventilator before death, which can be essential to correctly discern the effect of age on the expression of SARS-CoV2 entry factors as well as other genes. For example, stratifying patients on ventilator and not on ventilator is important, as it has been shown before that ACE2 expression may be crucial in recovering from ventilator injuries (D. Wang et al. 2019). Accounting for this metadata may lead to a different dependence between ACE2 expression in bulk lung data as another preprint shows (Santesmasses et al. 2020).

We thank the reviewer for the insightful comments. In this revision, we have now accounted for several additional covariates (including sex, smoking status, and ventilator status) in the analysis of entry factor expression, finding that age is weakly (but significantly) associated with ACE2 expression (new Figure 7; pg 11, lines 314-336).

3. The authors use a ‘gene signature based approach’ to identify cell type composition, which relies on the publicly available bulk RNAseq datasets obtained from cell lines or FACS sorted cell types not necessarily from the lung. Because of that we advise using a single-cell reference obtained from the lung to perform the cell type decomposition in the lung, which may lead to more accurate results. If that is not possible - it is at least worth mentioning the possible disadvantages of using ‘signature based’ as opposed to deconvolution approach in the discussion.

This a fantastic suggestion. We have now utilized the scRNA-seq datasets as a reference for inferring cell type compositions in the bulk lung transcriptomes (revised Figure 3; pg 5, lines 141-156). Similar to the xCell signature-based results, we find that epithelial cell types (such as AT2 and AT1 cells) and macrophages decrease in abundance with aging. Conversely, fibroblasts and vascular smooth muscle cells increase in abundance with aging. We have moved the xCell analyses to the supplementary figures, and have revised the text to describe the limitations of the xCell approach.

4. When describing change in T cell composition with age in lung the authors should cite extensive research work done on that topic and findings across various tissues, including lung (Thome et al. 2014; Kumar, Connors, and Farber 2018).

We appreciate the reviewer’s suggestions regarding the prior literature on aging-related changes in the T cell compartment. We have revised the manuscript accordingly (pg 7, lines 187-190).

Minor points:

1. Page 4, mention number of donors for lung samples in GTEx;

We have now done so (pg 3, line 82).

2. We advise that authors separate their findings into sections, and give them different subtitles so it is easier to comprehend the main findings, for example: (i) age-related changes in lung gene expression and cell type composition; (ii) cross-reference of genes changing with age in lung with the a) genes important for infection with SARS-CoV; b) genes responding and interacting with SARS-CoV2 genes.

We have added section headers to improve clarity of the manuscript.

3. Fig.2. Mention the number of genes in Muscle-enriched Cluster 1 genes and AT2-enriched Cluster 2 genes that was used to perform gene ontology analysis;

We have now added this information (pg 5, lines 128+131).

4. “On the transcriptional level, we first identified 1,285 genes that exhibit age-associated expression patterns.” - We suggest to delete “first” to avoid the inference that the authors were the first to study ageing in the GTEx data set, This analysis has been performed in the past with the aim to derive ageing gene expression signatures across multiple tissues, including lung (see, for example (Yang et al. 2015)). (The “first” may of course simply refer to the order in which the analysis was done.)

Our intention was indeed to refer to the order in which the analysis was done. Nevertheless, to avoid confusion, we have now revised the text accordingly (pg 12, line 342).

5. The authors also use different colour schemes to show the same data. A number of figures include heatmaps showing “% of expressing cells”. While the scales will vary with the data, the use of completely different colour schemes is confusing.

We apologize for the confusing color schemes. We have remade the heatmaps to use the same color schemes for a given type of data.

6. Throughout the paper, the authors use two different single-cell lung datasets (Tissue stability and Lung Cell Atlas) and infer cell enrichment using a collection of publicly available datasets within XCell package, which leads to different cell type annotation on different figures and initial confusion in understanding. Thus we advise authors clearly state the origin of cell annotations in the legend (where it is not present) and in the figure itself and emphasize it in the text (for cell type enrichment analysis with XCell).

We wholeheartedly agree with these suggestions, and apologize for the confusion. We have revised the text, figures, and legends accordingly to clearly label the reference dataset for a given figure. In the revised manuscript, all main figure panels are exclusively in reference to the Human Lung

Cell Atlas, while cell types from xCell and the Tissue Stability Cell Atlas are entirely in the supplementary figures.

References

- Kumar, Brahma V., Thomas J. Connors, and Donna L. Farber. 2018. "Human T Cell Development, Localization, and Function throughout Life." *Immunity* 48 (2): 202–13.
- Muus, Christoph, Malte D. Luecken, Gokcen Eraslan, Avinash Waghray, Graham Heimberg, Lisa Sikkema, Yoshihiko Kobayashi, et al. n.d. "Integrated Analyses of Single-Cell Atlases Reveal Age, Gender, and Smoking Status Associations with Cell Type-Specific Expression of Mediators of SARS-CoV-2 Viral Entry and Highlights Inflammatory Programs in Putative Target Cells." <https://doi.org/10.1101/2020.04.19.049254>.
- Santesmasses, Didac, José Pedro Castro, Aleksandr A. Zenin, Anastasia V. Shindyapina, Maxim V. Gerashchenko, Bohan Zhang, Csaba Kerepesi, Sun Hee Yim, Peter O. Fedichev, and Vadim N. Gladyshev. 2020. "COVID-19 Is an Emergent Disease of Aging." *Infectious Diseases (except HIV/AIDS)*. medRxiv. <https://doi.org/10.1101/2020.04.15.20060095>.
- Thome, Joseph J. C., Naomi Yudanin, Yoshiaki Ohmura, Masaru Kubota, Boris Grinshpun, Taheri Sathaliyawala, Tomoaki Kato, Harvey Lerner, Yufeng Shen, and Donna L. Farber. 2014. "Spatial Map of Human T Cell Compartmentalization and Maintenance over Decades of Life." *Cell* 159 (4): 814–28.
- Wang, Allen, Joshua Chiou, Olivier B. Poirion, Justin Buchanan, Michael J. Valdez, Jamie M. Verheyden, Xiaomeng Hou, et al. 2020. "Single Nucleus Multiomic Profiling Reveals Age-Dynamic Regulation of Host Genes Associated with SARS-CoV-2 Infection." bioRxiv. <https://doi.org/10.1101/2020.04.12.037580>.
- Wang, Di, Xiao-Qing Chai, Costan G. Magnussen, Graeme R. Zosky, Shu-Hua Shu, Xin Wei, and Shan-Shan Hu. 2019. "Renin-Angiotensin-System, a Potential Pharmacological Candidate, in Acute Respiratory Distress Syndrome during Mechanical Ventilation." *Pulmonary Pharmacology & Therapeutics* 58 (October): 101833.
- Yang, Jialiang, Tao Huang, Francesca Petralia, Quan Long, Bin Zhang, Carmen Argmann, Yong Zhao, et al. 2015. "Synchronized Age-Related Gene Expression Changes across Multiple Tissues in Human and the Link to Complex Diseases." *Scientific Reports* 5 (October): 15145.

Reviewed by Dr. Sarah Teichmann, Dr. Kerstin Meyer and Veronika Kedlian.

Reviewer #2 (Remarks to the Author):

This is a bioinformatics study that compares the aging transcriptome with siRNA screen of host factors involved in SARS-CoV2 infection and data from a SARS-CoV2 human protein-protein interaction map.

The results from analysis of changes in the aging lung are well done (Figs. 1-3) and reveals interesting new data. This includes the finding that the aging lung is characterized by increased vascular smooth muscle contraction, reduced mitochondrial activity, and decreased lipid metabolism. A particularly noteworthy finding is that expression of SARS-CoV2 entry factors (ACE2, TMPRSS2 and CTSL) are not significantly changed, at least at the mRNA level, supporting the notion that other host factors involving innate immunity may be more important. In this regard, the data showing changes in macro phage numbers and phenotype will be important to the field.

I am not sure the data with A549 cells and other lung cancer cells (Fig. 6) contribute much to our understanding, and relevance to conclusions are unclear.

We appreciate the reviewer’s comments on the manuscript. In our view, the data shown in the original Figure 6 can generate hypotheses into how age-related changes in the lung may be related to the direct transcriptional consequences of SARS-CoV-2 infection on lung cells. In this revision, we have also examined whether the transcriptional changes in severe COVID-19 patients are also enriched for age-associated genes (new Figure 6, below; pg 10, lines 285-312). We find that this was indeed the case, and we observed commonalities between the in vitro (with cancer cell lines) and in vivo (with severe COVID-19 patients) transcriptional changes induced by SARS-CoV-2 infection.

We nevertheless agree with the reviewer that interpretation of these findings is not straightforward. As we are hesitant to overinterpret our findings, we instead leave it to future studies to elucidate the functional relevance of these phenomena.

Reviewer #3 (Remarks to the Author):

In this study, the authors analysed published experimental data to determine how the interaction between SARS-CoV-2 and host cell may be regulated by age-associated factors. They postulate that these age-associated genes may explain COVID-19 pathogenesis in the elderly. Unfortunately, the conclusions drawn are not supported by the data presented.

(I) Part of the analysis was performed using published data on SARS-CoV and it is not justifiable to extrapolate this to SARS-CoV-2. As the authors have also stated “they are nevertheless two distinct viruses with different epidemiological features, indicating unique virology and host biology”. Indeed, these two viruses have been shown to have different viral kinetics in human infection and also significant difference in clinical outcome. Thus, it is not clear how the experimental data on SARS-CoV strengths this study.

We agree with the reviewer, and we have now moved all of the SARS-CoV data to the supplementary figures. We have also greatly cut down the description of these analyses in the main text (condensed to one paragraph) to deemphasize these analyses (pg 7, lines 192-206).

(II) The experimental data used in this study do not include any study obtained from COVID-19 patients. Rather it relied on interactomic and transcriptomic studies published by other groups in human cell lines. In addition, the authors did not determine if the observations from these profiling are replicated in clinical samples taken COVID-19 patients.

We appreciate the reviewer’s excellent suggestion. In this revision, we have now incorporated new analyses on transcriptional profiles from the lungs of COVID-19 patients. We find that age-associated genes are indeed dysregulated in COVID-19 patients, echoing the results from the in vitro cell models (see new Figure 6, below; pg 10, lines 285-312).

(III) Most importantly, none of the age-associated genes or pathway identified in this study was functionally tested in cell culture or animal model of infection. Thus, it is not possible to know if they have any contribution to COVID-19 pathogenesis in the elderly.

We completely agree with the reviewer that further functional studies are necessary to confirm the contribution of these age-associated changes to COVID-19 pathogenesis. The goal of the present manuscript is to explore potential associations between features of the aging lung and SARS-CoV-2/COVID-19. We hope to share these analyses to the field by generating a collection of testable hypotheses that await experimental validations. We have noted this clearly in the revised manuscript (for example, page 13, lines 365-371) and we have toned down the claims. We look forward to seeing whether experimental findings from other groups will confirm or refute the hypotheses we have set forth here.

REVIEWER COMMENTS

Reviewer #1 (Remarks to the Author):

We have read the author's response and consider that our previous comments have been addressed, but this leaves new questions open. Specifically, covariate influence on age-associated expression in SARS-CoV2 entry factors requires additional attention, detailed in major point 1 below.

Also after carefully rereading the updated manuscript, we have noted several new issues with the approaches used to evaluate age-associated gene expression, and performing cell type deconvolution/enrichment, which we detail in major points below.

Major points:

1. Effects of potential covariates (such as sex, BMI, ventilator status and others) have not been assessed on the age-related change in gene expression and cell type composition. We have previously raised the importance of including covariates to correctly discern the effect of age on the expression of the SARS-CoV2 entry factors, but including covariates is equally important to correctly assess the effect of age on the expression of all the genes and cell type composition (given that it has not been done previously).

In all the cases (calling differentially expressed genes with age & change in cell type composition) the authors need to think carefully about the covariates that should be included in the multiple regression model. Our advice on this analysis includes:

- a) which covariates have been shown to have an influence on gene expression (based on previous studies);
- b) perform exploratory analysis first to understand which covariates are correlated with age and, thus, can confound age effect;
- c) consider which covariates are correlated with age just by chance of the dataset and which ones are causally linked (i.e. aged patients are more likely to be placed on a ventilator) and thus, it may be appropriate to introduce interaction terms between the covariates.

The authors should then detail their decision process in a few words (and, perhaps, show the plots from the exploratory analysis in step b) in Supplementaries and correct the inference for the differentially expressed genes/cell types with age. The same argumentation has to be included for

the previously requested analysis of covariate influence on age-associated expression in SARS-CoV2 entry factors.

2. In the current version of the manuscript, authors present both cell type deconvolution with MuSIC and cell enrichment results performed with XCell on the bulk RNA lung transcriptomes from patients of different ages (taken from GTEx). Authors comment that both analyses show similar results, however, it is not possible to tell (except for a few matching cell types) given that both analyses have been performed using different cell type annotations (for example, with regard to macrophages and T cells). To solve that problem authors can perform XCell enrichment on the Human Lung Cell Atlas single-cell data to annotate cell types corresponding to XCell database ones.

After cell types are matched authors should highlight consistent and inconsistent results between two approaches.

3. Cell type frequencies in each sample that are produced by MuSIC bulk deconvolution is an example of compositional data, a type of data which is constrained by the total sum. That means that cell type proportions in each sample are negatively correlated (due to how proportions work) and it is impossible to tell if cell type proportion genuinely increased with age, or it is increased because another cell type decreased (see an example of cell type proportion analysis and review about methods for compositional data (Angelidis et al. 2019; Quinn et al. 2019)).

Consequently, we think that Kruskal-Wallis may not be correctly discerning the age-related change in cell type composition and other approaches (that can handle compositional data) should be used.

Minor points

1. Authors mention that overlap between age-associated genes, on one side, and in vitro or in vivo SARS-CoV2 induced genes, on the other side, are quite similar in terms of pathways and functional enrichment, which makes having two separate sections devoted to that in a main text a bit redundant.

Taken together with the fact that the current manuscript is quite large and has a total of 7 figures, we think it will make the understanding easier if 1) Figure 5 demonstrating comparison between age-associated genes and different in vitro evidence for genes activated in response to SARS-CoV2

changes will be moved to Supplementaries; 2) the results from two sections 'Age-associated genes are regulated by SARS-CoV-2 infection in vitro' and 'Age-associated genes are dysregulated in patients with severe COVID-19' will be merged.

2. Authors should provide the deconvolution results with MuSiC for all Human lung cell atlas (HLCA) cell types in Supplementaries, not only the significant ones (shown on Fig.3).

References

Angelidis, Ilias, Lukas M. Simon, Isis E. Fernandez, Maximilian Strunz, Christoph H. Mayr, Flavia R. Greiffo, George Tsitsiridis, et al. 2019. "An Atlas of the Aging Lung Mapped by Single Cell Transcriptomics and Deep Tissue Proteomics." *Nature Communications* 10 (1): 963.

Quinn, Thomas P., Ionas Erb, Greg Gloor, Cedric Notredame, Mark F. Richardson, and Tamsyn M. Crowley. 2019. "A Field Guide for the Compositional Analysis of Any-Omics Data." *GigaScience* 8 (9). <https://doi.org/10.1093/gigascience/giz107>.

Reviewed by Dr. Sarah Teichmann and PhD student Veronika Kedlian.

Reviewer #2 (Remarks to the Author):

The authors have satisfactorily addressed my concerns.

Reviewer #3 (Remarks to the Author):

The authors have added the analysis of a new dataset from a small study on COVID-19 patients. It seems to support their conclusions. Overall, the revised manuscript has addressed my previous concerns adequately.

REVIEWER COMMENTS

Reviewer #1 (Remarks to the Author):

We have read the author's response and consider that our previous comments have been addressed, but this leaves new questions open. Specifically, covariate influence on age-associated expression in SARS-CoV2 entry factors requires additional attention, detailed in major point 1 below.

Also after carefully rereading the updated manuscript, we have noted several new issues with the approaches used to evaluate age-associated gene expression, and performing cell type deconvolution/enrichment, which we detail in major points below.

Major points:

1. Effects of potential covariates (such as sex, BMI, ventilator status and others) have not been assessed on the age-related change in gene expression and cell type composition. We have previously raised the importance of including covariates to correctly discern the effect of age on the expression of the SARS-CoV2 entry factors, but including covariates is equally important to correctly assess the effect of age on the expression of all the genes and cell type composition (given that it has not been done previously).

In all the cases (calling differentially expressed genes with age & change in cell type composition) the authors need to think carefully about the covariates that should be included in the multiple regression model. Our advice on this analysis includes:

- a) which covariates have been shown to have an influence on gene expression (based on previous studies);
- b) perform exploratory analysis first to understand which covariates are correlated with age and, thus, can confound age effect;
- c) consider which covariates are correlated with age just by chance of the dataset and which ones are causally linked (i.e. aged patients are more likely to be placed on a ventilator) and thus, it may be appropriate to introduce interaction terms between the covariates.

The authors should then detail their decision process in a few words (and, perhaps, show the plots from the exploratory analysis in step b) in Supplementaries and correct the inference for the differentially expressed genes/cell types with age. The same argumentation has to be included for the previously requested analysis of covariate influence on age-associated expression in SARS-CoV2 entry factors.

We appreciate the reviewer's insightful comments and helpful suggestions. In light of their advice, we have performed new analyses to investigate the relationship between age and other variables in the GTEx dataset. In regards to the three bullet points described above:

a) There is literature indicating that age¹⁻³, sex⁴, obesity⁵, hypertension⁶, diabetes⁷, and smoking⁸ are all associated with changes in gene expression (many more studies not cited here). As the reviewer pointed out in the first round of review, ventilator status can also be associated with alterations in gene expression in the lung. We therefore proceeded to consider all of these potential covariates (see below).

b and c) We fit a log-linear model on the full contingency table (Poisson regression), further considering all pairwise interaction terms that involve age (new **Supplementary Figure 2**, below; details of model in Table S41). This model was well-fitted to the observed counts (residual deviance = 112.76 on 145 degrees of freedom; deviance Chi-square $p = 0.978$).

In the figure above, a coefficient > 0 indicates that the clinical feature (or feature pair) is associated with increased counts in the GTEx cohort. When examining interaction terms involving age, we find that only the interaction between age and hypertension is statistically significant. The magnitude of this interaction is positive and progressively increases with age: 30-39 (coefficient = 0.71), 40-49 (1.55), 50-59 (1.95), 60-69 (1.95), and 70-79 (2.07). This indicates that within the GTEx cohort, hypertension is more common in older age groups (such that the estimated “effect” of hypertension on patient frequency is increasingly positive with older age). This is consistent with the well-established relationship between aging and hypertension⁹⁻¹¹, benchmarking the validity of the statistical approach.

While the reviewer has suggested that the age-association of hypertension status should subsequently be controlled for when performing differential expression analysis, we would instead argue the opposite. On a mechanistic level, the physiologic changes that occur with aging (such as increased arterial stiffness and vascular inflammation) are thought to contribute to the development of hypertension. Aging can be defined as the “time-related deterioration of the physiological functions necessary for survival and fertility”¹²; seen from this perspective, hypertension can therefore be thought of as a disease of “cardiovascular aging.”¹³

Furthermore, we note that the low resolution of the clinical annotations in GTEx precludes a functional understanding of the hypertension “yes/no” label. Based on the information that was accessible to us, it is unknown for how many years a given donor had been diagnosed with hypertension, the severity of the hypertension, what medications they took (if any), and whether their hypertension was well-controlled. None of these important factors can be gleaned from the simple “yes/no” data provided in GTEx, and yet these features are all crucial for understanding the pathophysiological consequences of hypertension in the clinical setting.

For these reasons, we do not find it appropriate to control for hypertension as a covariate in this context. Not only is hypertension itself a disease of aging, the binary classification of hypertension in GTEx belies a wealth of diversity in clinical severity, chronicity, management, and associated outcomes. Nevertheless, we again thank the reviewer for their insightful comments. We have now detailed the new analyses and our subsequent decision process in the manuscript text.

2. In the current version of the manuscript, authors present both cell type deconvolution with MuSIC and cell enrichment results performed with XCell on the bulk RNA lung transcriptomes from patients of different ages (taken from GTEx). Authors comment that both analyses show similar results, however, it is not possible to tell (except for a few matching cell types) given that both analyses have been performed using different cell type annotations (for example, with regard to macrophages and T cells). To solve that problem authors can perform XCell enrichment on the Human Lung Cell Atlas single-cell data to annotate cell types corresponding to XCell database ones. After cell types are matched authors should highlight consistent and inconsistent results between two approaches.

This is an excellent suggestion, though we note that xCell was designed for bulk transcriptomic data, so it is not entirely suitable for analyzing scRNA-seq data. Fortunately, the authors of xCell have since created SingleR, which is similar to xCell but designed for scRNA-seq data. While not all of the original xCell cell types are also annotated in SingleR, 43 out of 64 xCell annotations

were perfectly matched in SingleR. We therefore used SingleR to re-annotate the Human Lung Cell Atlas (pre-processed to pseudo-bulk transcriptomes for each HLCA cell type).

By cross-referencing the two sets of cell type annotations, we found a moderate degree of consistency between the two annotations (see **new Supplementary Figure 5d**, below). For instance, all of the epithelial cell types in the HLCA were successfully classified as “Epithelial cells” by SingleR. However, there were several clear-cut cases of misclassification. As an example, HLCA alveolar fibroblasts and vascular smooth muscle cells were both annotated as adipocytes by SingleR.

The number of discordant cell type annotations lowers our overall confidence in the xCell-based analyses, and accordingly, we have significantly cut down the section of the manuscript where we describe these results. As we discuss in the text, this discrepancy likely reflects a mismatch between the human lung and the datasets that were used by SingleR to construct the cell type enrichment scores. However, we feel it is still reasonable to conclude that, at least to some extent, the xCell and MuSiC approaches found consistent results (e.g., an age-associated decreases in epithelial cells and macrophages).

3. Cell type frequencies in each sample that are produced by MuSiC bulk deconvolution is an example of compositional data, a type of data which is constrained by the total sum. That means that cell type proportions in each sample are negatively correlated (due to how proportions work) and it is impossible to tell if cell type proportion genuinely increased with age, or it is increased because another cell type decreased (see an example of cell type proportion analysis and review about methods for compositional data (Angelidis et al. 2019; Quinn et al. 2019)).

Consequently, we think that Kruskal-Wallis may not be correctly discerning the age-related change in cell type composition and other approaches (that can handle compositional data) should be used.

We thank the reviewer for the insightful comments and advice. We have now performed differential proportionality analysis using propr¹⁴. As described in the review by Quinn et al. that the reviewer helpfully suggested, propr analysis seeks to identify changes in feature “stoichiometry” – when applied to the MuSiC proportion estimates, propr examines whether the proportions of two different cell types change with respect to each other, as a function of a grouping label (in this case, age). From this analysis, we found several pairs of cell types whose relative proportions were significantly different as a function of age (see below, **new Supplementary Figure 4**). In this heatmap, only the theta values for cell type pairs with $\theta < 0.95$ were included to highlight the high-confidence pairings; theta values can range from 0 to 1, with lower values indicating a stronger difference between age groups.

We observed that the propr results were consistent with those identified from the Kruskal-Wallis statistics, in the sense that propr identified cell type pairs which were each composed of a cell type that increased with aging (by the KW test) and a cell type that decreased with aging. For instance, the strongest cell type pair was AT2 cells : alveolar fibroblasts (F-statistic adj. $p = 7.47E-43$), the 2nd strongest pair was AT2 cells : vascular smooth muscle cells (adj. $p = 8.31E-42$), and the 3rd strongest pair was AT1 cells : alveolar fibroblasts (adj. $p = 1.70E-40$). These pairings are consistent with our expectations from the Kruskal-Wallis analysis, which had found that AT1 and AT2 cells decreased in proportion with age, while alveolar fibroblasts and vascular smooth muscle cells increased in proportion with age.

Given the complementary conclusions between these different statistical approaches, we feel that it is appropriate to retain the analyses using Kruskal-Wallis test, as we feel these results are simpler for readers to interpret. Furthermore, we note that the non-parametric nature of the Kruskal-Wallis statistic means that it relies on very few assumptions about the underlying data; the test is comparing the *ranks* of cell type proportions across age groups, not their absolute abundances.

In short, we have now included the new differential proportionality analysis in **Supplementary Figure 4**, but we have opted to retain the original Kruskal-Wallis analysis in the main Figure 3. As discussed above, the two types of analyses led to conceptually similar conclusions, but we feel the Kruskal-Wallis analysis is more intuitive to understand. Nevertheless, we have revised the manuscript throughout to emphasize that MuSiC estimates are proportions, as well as the importance of using statistical methods that are designed for compositional data. We greatly appreciate the reviewer for illuminating the complexities intrinsic to this type of analysis.

Minor points

1. Authors mention that overlap between age-associated genes, on one side, and in vitro or in vivo SARS-CoV2 induced genes, on the other side, are quite similar in terms of pathways and functional enrichment, which makes having two separate sections devoted to that in a main text a bit redundant.

Taken together with the fact that the current manuscript is quite large and has a total of 7 figures, we think it will make the understanding easier if 1) Figure 5 demonstrating comparison between age-associated genes and different in vitro evidence for genes activated in response to SARS-CoV2 changes will be moved to Supplementaries; 2) the results from two sections ‘Age-associated genes are regulated by SARS-CoV-2 infection in vitro’ and ‘Age-associated genes are dysregulated in patients with severe COVID-19’ will be merged.

We appreciate the reviewer's suggestion. However, we believe it is pertinent to distinguish gene expression changes observed *in vitro* from those *in vivo*. In particular, the *in vitro* data are derived from pure populations of lung cancer cell lines, while the *in vivo* data draw from a mixture of different lung cell types. Thus, we would argue that these two comparisons offer distinct but complementary insights into the relationship between the gene expression changes observed with SARS-CoV-2/COVID-19 and aging.

With that said, we have moved the latter half of Figure 5 to Supplementary Figure 9, since the original Figure 5 was rather dense and had too many subpanels. We also agree that having 2 distinct section headers for *in vitro* and *in vivo* is somewhat redundant, so we have now merged them.

Additionally, we have moved the former Figure 7 (regression analysis on entry factor expression) to the supplementary figures, as we feel that this analysis is thematically and methodologically distinct from the other main figures. We hope the reviewer agrees that the current version of the manuscript (6 main figures, 14 supplementary figures) is structured in such a way that improves the logical flow and facilitates ease of understanding.

2. Authors should provide the deconvolution results with MuSiC for all Human lung cell atlas (HLCA) cell types in Supplementaries, not only the significant ones (shown on Fig.3).

We have now done so (see new **Supplementary Figure 5d**, and shown above in the response to major comment #2). We also note that the scaled MuSiC proportions have been provided in the Supplementary Tables.

References

1. Yang, J. *et al.* Synchronized age-related gene expression changes across multiple tissues in human and the link to complex diseases. *Scientific Reports* **5**, 15145 (2015).
2. Kedlian, V. R., Donertas, H. M. & Thornton, J. M. The widespread increase in inter-individual variability of gene expression in the human brain with age. *Aging (Albany NY)* **11**, 2253–2280 (2019).
3. Angelidis, I. *et al.* An atlas of the aging lung mapped by single cell transcriptomics and deep tissue proteomics. *Nat Commun* **10**, 1–17 (2019).
4. Gershoni, M. & Pietrokovski, S. The landscape of sex-differential transcriptome and its consequent selection in human adults. *BMC Biology* **15**, 7 (2017).
5. Hao, R.-H. *et al.* Gene expression profiles indicate tissue-specific obesity regulation changes and strong obesity relevant tissues. *International Journal of Obesity* **42**, 363–369 (2018).
6. Huan, T. *et al.* A meta-analysis of gene expression signatures of blood pressure and hypertension. *PLoS Genet.* **11**, e1005035 (2015).
7. Planas, R., Pujol-Borrell, R. & Vives-Pi, M. Global gene expression changes in type 1 diabetes: insights into autoimmune response in the target organ and in the periphery. *Immunol. Lett.* **133**, 55–61 (2010).
8. Brody, J. S. Transcriptome alterations induced by cigarette smoke. *International Journal of Cancer* **131**, 2754–2762 (2012).
9. Buford, T. W. Hypertension and Aging. *Ageing Res Rev* **26**, 96–111 (2016).

10. Mozaffarian, D. *et al.* Heart disease and stroke statistics--2015 update: a report from the American Heart Association. *Circulation* **131**, e29-322 (2015).
11. Ong Kwok Leung, Cheung Bernard M.Y., Man Yu Bun, Lau Chu Pak & Lam Karen S.L. Prevalence, Awareness, Treatment, and Control of Hypertension Among United States Adults 1999–2004. *Hypertension* **49**, 69–75 (2007).
12. Gilbert, S. F. Aging: The Biology of Senescence. *Developmental Biology. 6th edition* (2000).
13. Sun Zhongjie. Aging, Arterial Stiffness, and Hypertension. *Hypertension* **65**, 252–256 (2015).
14. Quinn, T. P., Richardson, M. F., Lovell, D. & Crowley, T. M. propr: An R-package for Identifying Proportionally Abundant Features Using Compositional Data Analysis. *Scientific Reports* **7**, 16252 (2017).

Reviewed by Dr. Sarah Teichmann and PhD student Veronika Kedlian.

Reviewer #2 (Remarks to the Author):

The authors have satisfactorily addressed my concerns.

We thank the reviewer for their helpful suggestions in reviewing our manuscript.

Reviewer #3 (Remarks to the Author):

The authors have added the analysis of a new dataset from a small study on COVID-19 patients. It seems to support their conclusions. Overall, the revised manuscript has addressed my previous concerns adequately.

We thank the reviewer for their helpful suggestions in reviewing our manuscript.

REVIEWER COMMENTS

Reviewer #1 (Remarks to the Author):

We have read through the author's response to our comments and consider that our comments 2 and 3 have been addressed in full. However, the author's reply to point 1 requires more clarification and additional work. We also noted two new issues which arise from the author's replies to our comments.

1. Authors indicate that there is substantial evidence for sex, obesity, hypertension, diabetes and smoking to be associated with changes in gene expression together with age. We fully agree that some of the covariates here, such as hypertension and T2D can be caused by ageing and, thus, shouldn't be corrected for as they capture ageing gene expression signature. However, we think that authors at the very least should include sex and smoking status, which are not causally related with age, have a major influence on the transcriptome and frequently used as covariates in similar ageing gene expression studies.

2. When trying to assess effect of age on gene expression authors didn't consider at all ischemic time, hardy scale and autolysis score, which serve as indicators of sample quality (level of RNA degradation etc) and have a dramatic influence on cell transcriptome. This can be obvious from the attached plot indicating a correlation between PC1 and ischemic time generated by us on GTEX website (see attachment). We strongly advise authors to include one of the following covariates (give that they are correlated) in the model.

3. Authors mention that the proportion of macrophages decreases with age according to both MUSIC and XCell approaches. However, according to MUSIC cell type deconvolution, it is only the population of proliferating macrophages that decreases with age, unlike the whole macrophage cluster in XCell. This population of proliferating macrophages likely constitutes a very small proportion of the macrophage cluster and is dominated by a proliferation signature. Decrease in this population can reflect a decrease in cell proliferation with age, but not genuine change in the macrophage population. Thus, we would advise against extrapolating inference on this population to the whole macrophage cluster. Also, as authors concluded xCell annotations do not match very well with HLCA cell types and misclassify a lot of other cell types as macrophages, which casts doubt on the macrophage decrease evidence from XCell. Consequently, authors should tone down the claims regarding the decrease in macrophage population throughout the paper.

Minor points:

1. We are not sure whether the log-linear model on the full contingency table used by the authors to investigate if there is a confounding effect between age and covariates is correctly addressing the given question. Covariates in these types of models are supposed to have an additive predictive effect which doesn't quite make sense in this particular occasion. In our opinion, it is best to analyse confounding between age and other factors by investigating each factor one by one and, possibly, treating age as a continuous covariate rather than categorical one. Thus, we would like to have more explanation for why this analysis addresses the question that authors ask (possibly authors also can get external statistical opinion on this issue) or we would recommend removing this analysis from the paper.

Reviewed by Dr Sarah Teichmann and PhD student Veronika Kedlian.

REVIEWER COMMENTS

Reviewer #1 (Remarks to the Author):

We have read through the author's response to our comments and consider that our comments 2 and 3 have been addressed in full. However, the author's reply to point 1 requires more clarification and additional work. We also noted two new issues which arise from the author's replies to our comments.

We thank the reviewer for reading the manuscript again and providing additional insightful suggestions. We have performed additional work and revised the manuscript accordingly. To facilitate review, the updated parts are highlighted in the revised manuscript.

1. Authors indicate that there is substantial evidence for sex, obesity, hypertension, diabetes and smoking to be associated with changes in gene expression together with age. We fully agree that some of the covariates here, such as hypertension and T2D can be caused by ageing and, thus, shouldn't be corrected for as they capture ageing gene expression signature. However, we think that authors at the very least should include sex and smoking status, which are not causally related with age, have a major influence on the transcriptome and frequently used as covariates in similar ageing gene expression studies.

As suggested by the reviewer, we have now re-performed all of the analyses to control for sex and smoking status, as well as Hardy scale (see response to point 2 below).

2. When trying to assess effect of age on gene expression authors didn't consider at all ischemic time, hardy scale and autolysis score, which serve as indicators of sample quality (level of RNA degradation etc) and have a dramatic influence on cell transcriptome. This can be obvious from the attached plot indicating a correlation between PC1 and ischemic time generated by us on GTEX website (see attachment). We strongly advise authors to include one of the following covariates (give that they are correlated) in the model.

While we are not able to view the attachment and thus cannot comment on it, we appreciate the reviewer pointing out the importance of factors such as ischemic time, Hardy scale, and autolysis score on the GTEX data. As suggested by the reviewer, we have now re-performed all of the analyses to account for sex, smoking status, and Hardy scale. The revised analysis similarly revealed hundreds of age-associated genes/transcripts. The revised gene set was distinct from our prior gene set, though the two significantly overlapped (see below). We subsequently used the revised set of age-associated genes to carry out all downstream analyses, which we present in the revised manuscript.

3. Authors mention that the proportion of macrophages decreases with age according to both MUSIC and XCell approaches. However, according to MUSIC cell type deconvolution, it is only the population of proliferating macrophages that decreases with age, unlike the whole macrophage cluster in XCell. This population of proliferating macrophages likely constitutes a very small proportion of the macrophage cluster and is dominated by a proliferation signature. Decrease in this population can reflect a decrease in cell proliferation with age, but not genuine change in the macrophage population. Thus, we would advise against extrapolating inference on this population to the whole macrophage cluster. Also, as authors concluded xCell annotations do not match very well with HLCA cell types and misclassify a lot of other cell types as macrophages, which casts doubt on the macrophage decrease evidence from XCell. Consequently, authors should tone down the claims regarding the decrease in macrophage population throughout the paper.

We thank the reviewer for the detailed suggestions. In the revised manuscript, we now use CIBERSORTx instead of MuSiC for bulk deconvolution. CIBERSORTx similarly utilizes a single cell transcriptome reference to deconvolute bulk transcriptome profiles, and has been reported to have better performance over MuSiC. The CIBERSORTx-based analysis did not reveal any significant changes in macrophages (proliferative or otherwise; see **new Figure 3**, below). Accordingly, we have removed all statements regarding age-associated changes in macrophages in the revised manuscript.

We would also like to highlight that these new analyses on cell composition are now fully accounting for sex, smoking status, and Hardy scale, as suggested by the reviewer in prior comments. Since neither the Kruskal-Wallis test nor the propr differential proportionality analysis that we had used previously are able to account for additional confounding variables, the revised analysis instead uses a proportional odds ordinal logistic regression (OLR) model. OLR is in fact a generalization of the Kruskal-Wallis test, with the key difference that it is able to accommodate multifactorial designs (OLR is similarly non-parametric). Finally, given the several limitations with the xCell-based annotations, we have chosen to entirely remove these analyses from the revised manuscript.

Minor points:

1. We are not sure whether the log-linear model on the full contingency table used by the authors to investigate if there is a confounding effect between age and covariates is correctly addressing the given question. Covariates in these types of models are supposed to have an additive predictive effect which doesn't quite make sense in this particular occasion. In our opinion, it is best to analyse confounding between age and other factors by investigating each factor one by one and, possibly, treating age as a continuous covariate rather than categorical one. Thus, we would like to have more explanation for why this analysis addresses the question that authors ask (possibly authors also can get external statistical opinion on this issue) or we would recommend removing this analysis from the paper.

The log-linear model is assessing whether there is overrepresentation of certain covariates (such as hypertension or ventilator use) among older age groups, simply in regards to the number of samples. In light of the reviewer's recent comments, we agree that while this analysis is informative, it is tangential to the revised manuscript. We have removed this figure from the manuscript.

Reviewed by Dr Sarah Teichmann and PhD student Veronika Kedlian.

We greatly appreciate all of the reviewers' careful reading and thoughtful suggestions that helped us improved the manuscript.

REVIEWER COMMENTS

Reviewer #1 (Remarks to the Author):

We have read the author's response to our previous comments and consider that they have been addressed in full.

However, the author's reply to our comment about bulk deconvolution has opened a few new issues that we detail below.

Major comments

1. In the previous version of manuscript authors performed bulk deconvolution using MuSiC, while in this one authors switched to CybersortX by saying that CybersortX “has been reported to have better performance over MuSiC”. As we are not aware of such evidence, we kindly ask authors to provide a reference or more detailed explanation of that. Otherwise, given that cell type deconvolution results presented in the current manuscript differ from the previous one, it would be reasonable for authors to compare the results from both approaches and report the common differentially abundant cell types with age.

2. In the paper authors present deconvolution of bulk transcriptomes into 40 lung cell types. While this is technically possible, from our experience not all the results can be equally reliable: proportion of abundant cell types can be estimated more reliably than the rare ones. To that end, we suggest authors rank cell types by their prevalence (or count) in single-cell data and provide these counts (or proportions) side by side with the ageing change, possibly also tone down the claims for the rare populations.

3. In follow up to the last comment, we suggest that authors also prefilter cell types based on percentage of samples with non-zero proportion (we can advise to have at least 50% of non-zero samples for cell type) before proceeding with ageing analysis. Otherwise, the estimates of ageing changes are unreliable and can be driven by the several outlier samples as we have observed for type 2 myeloid dendritic cells & proliferating NK/T cells for which authors report that they decrease with age. We identified this issue by plotting the raw attached data as suggested in our minor comment 1 (see attached plots).

Minor comment

1. We would like to ask authors to provide boxplots of the estimated proportions vs age (before & after correction for confounding variables) for each cell type in

Supplementaries. From our experience seeing raw data helps with judging reliability of the conclusions.

Reviewed by Dr. Sarah Teichmann and PhD student Veronika Kedlian.

Reviewer #1 (Remarks to the Author):

We have read the author's response to our previous comments and consider that they have been addressed in full. However, the author's reply to our comment about bulk deconvolution has opened a few new issues that we detail below.

Major comments

1. In the previous version of manuscript authors performed bulk deconvolution using MuSiC, while in this one authors switched to CybersortX by saying that CybersortX “has been reported to have better performance over MuSiC”. As we are not aware of such evidence, we kindly ask authors to provide a reference or more detailed explanation of that. Otherwise, given that cell type deconvolution results presented in the current manuscript differ from the previous one, it would be reasonable for authors to compare the results from both approaches and report the common differentially abundant cell types with age.

We are aware of a handful of studies that directly compared CIBERSORTx to MuSiC, such as Jew et al. Nature Communications 2020 (see Figure 3a-c), and Menden et al. Science Advances 2020 (see Figure 3a-b). Nevertheless, the two algorithms likely have varying relative performance in different contexts. In order to more specifically evaluate these two approaches for analysis of GTEx data, we had also performed an internal benchmark using GTEx pancreas RNA-seq samples. This analysis revealed that CIBERSORTx was better able to recover the known loss of beta cells in patients with type 1 diabetes (CIBERSORTx: $p = 4.58E-05$, MuSiC: $p = 0.175776$). Thus, we proceeded with CIBERSORTx for our analyses.

2. In the paper authors present deconvolution of bulk transcriptomes into 40 lung cell types. While this is technically possible, from our experience not all the results can be equally reliable: proportion of abundant cell types can be estimated more reliably than the rare ones. To that end, we suggest authors rank cell types by their prevalence (or count) in single-cell data and provide these counts (or proportions) side by side with the ageing change, possibly also tone down the claims for the rare populations.

The reviewer suggested that we show the prevalence of the different cell types in the single-cell data. However, such an analysis would be inaccurate and misleading, because the Human Lung Cell Atlas data was actually generated by pre-enriching for different cell compartments prior to sequencing. From Travaglini et al.: “Lung samples were independently dissociated into single cell suspensions, and each lung cell suspension was then separated into epithelial (EPCAM+), endothelial/immune (CD31+/CD45+) and stromal (EPCAM-, CD31-/CD45-) populations by fluorescence-activated cell sorting (FACS) or magnetic-assisted cell sorting (MACS)”. Thus, it would not be accurate to show the relative abundances of the different cell types as captured in the single-cell data.

Instead, we now show the estimated proportions in the bulk RNA-seq lung samples (see **Supplementary Figure 5a**, below).

3. In follow up to the last comment, we suggest that authors also prefilter cell types based on percentage of samples with non-zero proportion (we can advise to have at least 50% of non-zero samples for cell type) before proceeding with ageing analysis. Otherwise, the estimates of ageing changes are unreliable and can be driven by the several outlier samples as we have observed for type 2 myeloid dendritic cells & proliferating NK/T cells for which authors report that they decrease with age. We identified this issue by plotting the raw attached data as suggested in our minor comment 1 (see attached plots).

For the previous manuscript submission, we had used the analysis results from the CIBERSORTx web-interface, as it was most readily available for use. However, one important limitation is that the web-interface is very limited in its processing power and memory limits, so we were forced to downsize the size of the scRNA-seq dataset (we had subsetted on the 8000 most variable genes). While this would not significantly affect the generation of the signature matrix (since highly variable genes are most likely to be incorporated into the signature matrix), it could however have an effect on the batch correction performed by CIBERSORTx. This is because CIBERSORTx compares the transcriptomes of scRNA-seq to bulk RNA-seq in order to assess systematic biases in the data due to technological differences. Thus, using the complete scRNA-seq dataset is important for accurate batch correction and subsequent deconvolution.

To this end, we have now reanalyzed the GTEx lung data with a local version of CIBERSORTx, using the full scRNA-seq matrix (no variance-based prefiltering of genes). In this version of the analysis, we observe that 8 cell types were absent in at least 50% of the samples (i.e. estimated proportion = 0; see below, **Supplementary Figure 5b**). These cell types were bronchial vessel 2, intermediate monocyte, basophil/mast 2, capillary intermediate 2, capillary, natural killer, myeloid dendritic type 2, and proliferating macrophage. As suggested by the reviewer, we subsequently excluded these cell types from further analysis. The revised analysis of age-associated cell types can be found in **Figure 3**.

Minor comment

1. We would like to ask authors to provide boxplots of the estimated proportions vs age (before & after correction for confounding variables) for each cell type in Supplementaries. From our experience seeing raw data helps with judging reliability of the conclusions.

We have now done so. Please see **Supplementary Figure 6** (unadjusted proportions; below), and **Supplementary Figure 7** (adjusted proportions, i.e. the residuals from a generalized linear model omitting age). We note that these plots are with binned ages, whereas our OLR model considers age as a continuous variable; thus, statistical significance in the OLR model may not always be readily appreciated from visual inspection of the boxplots.

REVIEWERS' COMMENTS

Reviewer #1 (Remarks to the Author):

We appreciate that the authors have performed extensive additional analyses to address our comments and can confirm that all our comments are addressed now in full.

Reviewed by Dr Sarah Teichmann and PhD student Veronika Kedlian

REVIEWERS' COMMENTS

Reviewer #1 (Remarks to the Author):

We appreciate that the authors have performed extensive additional analyses to address our comments and can confirm that all our comments are addressed now in full.

Reviewed by Dr Sarah Teichmann and PhD student Veronika Kedlian

We thank the reviewers again for their insights and suggestions throughout the review process.